# A secreted *Ustilago maydis* effector promotes virulence by targeting anthocyanin biosynthesis in maize

Shigeyuki Tanaka[1], Thomas Brefort[1†a], Nina Neidig[1], Armin Djamei[1†b], Jörg Kahnt[2], Wilfred Vermerris[3,4], Stefanie Koenig[5], Kirstin Feussner[5], Ivo Feussner[5], Regine Kahmann[1*]

[1]Department of Organismic Interactions, Max Planck Institute for Terrestrial Microbiology, Marburg, Germany; [2]Department of Ecophysiology, Max Planck Institute for Terrestrial Microbiology, Marburg, Germany; [3]Department of Microbiology & Cell Science, University of Florida, Gainesville, United States; [4]Genetics Institute, University of Florida, Gainesville, United States; [5]Albrecht-von-Haller-Institute, Georg-August-University Göttingen, Göttingen, Germany

**Abstract** The biotrophic fungus *Ustilago maydis* causes smut disease in maize with characteristic tumor formation and anthocyanin induction. Here, we show that anthocyanin biosynthesis is induced by the virulence promoting secreted effector protein Tin2. Tin2 protein functions inside plant cells where it interacts with maize protein kinase ZmTTK1. Tin2 masks a ubiquitin–proteasome degradation motif in ZmTTK1, thus stabilizing the active kinase. Active ZmTTK1 controls activation of genes in the anthocyanin biosynthesis pathway. Without Tin2, enhanced lignin biosynthesis is observed in infected tissue and vascular bundles show strong lignification. This is presumably limiting access of fungal hyphae to nutrients needed for massive proliferation. Consistent with this assertion, we observe that maize *brown midrib* mutants affected in lignin biosynthesis are hypersensitive to *U. maydis* infection. We speculate that Tin2 rewires metabolites into the anthocyanin pathway to lower their availability for other defense responses.

**\*For correspondence:**
kahmann@mpi-marburg.mpg.de

**†Present address:** [a]Comprehensive Biomarker Center, Heidelberg, Germany; [b]Gregor Mendel Institute of Molecular Plant Biology, Vienna, Austria

**Competing interests:** The authors declare that no competing interests exist.

**Reviewing editor**: Daniel J Kliebenstein, University of California, Davis, United States

## Introduction

The fungus *Ustilago maydis* causes smut disease in maize. *U. maydis* is a biotrophic fungus that needs living plant tissue for completion of its sexual cycle and spore formation. After penetrating epidermal cells and establishing an extended interaction zone in which hyphae are encased by the host plasma membrane, fungal hyphae traverse mesophyll tissue (*Doehlemann et al., 2008*, *2009*), and accumulate in and around vascular bundles (*Doehlemann et al., 2008*) presumably to obtain nutrients from veins. During infection, *U. maydis* secretes hundreds of effector proteins to suppress plant defense responses, and to reprogram plant signaling and metabolism (*Doehlemann et al., 2008*; *Mueller et al., 2008*; *Horst et al., 2010*; *Djamei and Kahmann, 2012*). However, the function of the vast majority of these novel effectors is unknown (*Djamei et al., 2011*; *Djamei and Kahmann, 2012*; *Hemetsberger et al., 2012*; *van der Linde et al., 2012*; *Mueller et al., 2013*). At later stages of host colonization, large plant tumors develop that provide the environment for massive fungal proliferation. During biotrophic growth, a high affinity fungal sucrose H$^+$ symporter, Srt1, is upregulated and plays a crucial role for fungal nutrition (*Wahl et al., 2010*).

Visible changes in host metabolism include an accumulation of anthocyanin in infected areas (*Banuett and Herskowitz, 1996*; *Brefort et al., in press*). Up to date, anthocyanin induction was thought to result from unspecific biotic stress caused by microbial infection (*Steyn et al., 2002*),

**eLife digest** The production of agricultural crop plants is severely hindered by bacteria, viruses, and fungi that have developed their own strategies to colonize these plants and obtain nutrients from them. Some pathogens kill the plants they colonize, but 'biotrophic pathogens' employ sophisticated strategies to manipulate the host plant without killing it.

During the past decade it has been recognized that the interactions between plants and biotrophic pathogens are largely governed by effector proteins—which are typically secreted by the pathogen after it makes contact with the host. These effector proteins can either stay in the space between the plant cells and pathogen cells, or actually enter inside the plant cells.

The fungus *Ustilago maydis* is one such biotrophic pathogen that colonizes maize plants and causes a disease called corn smut. Hallmarks of this infection are the formation of large plant tumors and the production of a red pigment, called anthocyanin, in infected plant tissues.

Now, Tanaka et al. reveal that an effector called Tin2, which is secreted by the corn smut fungus, causes the production of this anthocyanin pigment. Tin2 moves inside plant cells, where it blocks the breakdown of a protein-modifying enzyme that is necessary to 'switch on' the production of anthocyanin. When a mutant fungus that lacks Tin2 infects a maize plant, no anthocyanin is induced and the pathogen fails to reach the vascular tissue, where it would normally get most of its nutrients. Tanaka et al. revealed that, in these infections, this vascular tissue was strongly reinforced with a compound, called lignin, suggesting that the plant fortifies these cell walls to block access by the mutant fungus.

Since the building blocks for making lignin are also required for making anthocyanins, Tanaka et al. suggest a model whereby Tin2 compromises the ability of plants to protect themselves by diverting resources away from making lignin. In line with this speculation, the corn smut fungus was shown to cause stronger disease symptoms in maize plants with mutations that prevent them from producing lignin.

The Tin2 effector of the corn smut fungus appears to target a critical protein in maize that can shift the balance of the plant's metabolic pathways in favor of the pathogen. Further, since anthocyanin production is also observed after infections of plants with other microbes, these findings may have uncovered a microbial strategy to enhance virulence that is also employed by other plant pathogens.

and flavonoid-pathway related compounds like luteolinidin have been shown to function as phytoalexins negatively affecting fungal growth (*Snyder and Nicholson, 1990*; *Zuther et al., 2012*). So far, no positive link has been established between anthocyanin induction and microbial virulence. We have recently shown that the secreted *U. maydis* effector Tin2 (Um05302) is specifically responsible for anthocyanin induction in infected seedling tissue (*Brefort et al., in press*). The *tin2* effector gene resides in the largest effector gene cluster 19A comprising 24 effectors (*Kämper et al., 2006*; *Brefort et al., in press*). Deletion of the entire cluster 19A largely abolishes tumor formation and anthocyanin induction, although the mutant is still able to grow and complete its life cycle inside the plant tissue (*Brefort et al., in press*). Here, we elucidate the molecular function of Tin2 and show that Tin2 is a secreted anthocyanin biosynthesis inducing effector protein functioning inside plant cells. We hypothesize that it may be beneficial for *U. maydis* to induce anthocyanin biosynthesis as a strategy to compete with tissue lignification and to gain access to veins.

## Results

### *tin2* mutant phenotype

Tin2 was previously shown to be responsible for anthocyanin induction during biotrophic growth and was shown to have a weak contribution to virulence (*Brefort et al., in press*). To unambiguously establish its contribution to virulence, we introduced *tin2* in this study into SG200Δ19A-1 (*Brefort et al., in press*), a mutant strain lacking the 14 leftmost effector genes of cluster 19A, including *tin2*. This mutant is severely attenuated in virulence, is unable to elicit anthocyanin formation and induces at most small tumors (*Brefort et al., in press*). Introduction of *tin2* enhanced tumor formation significantly and fully restored anthocyanin biosynthesis (*Figure 1A,B*). In addition, we could demonstrate that

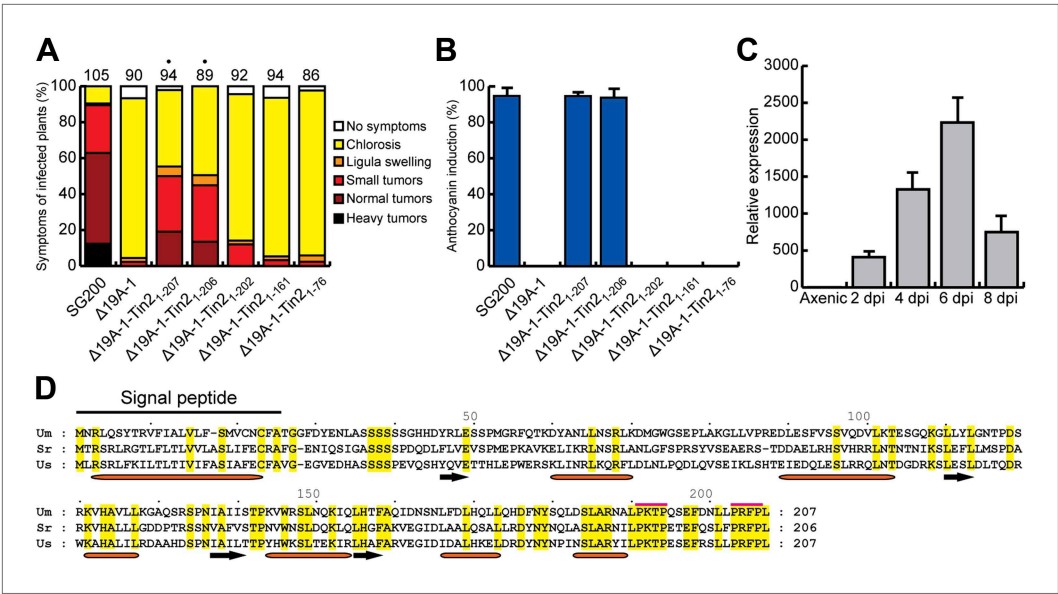

**Figure 1**. Expression and mapping of functional domains in Tin2. (**A**) Biological function of truncated Tin2 proteins. Virulence of SG200Δ19A-1 strains expressing either wild-type or C-terminally truncated Tin2 proteins. Numbers indicate total infected plants. Dots indicate that at least one of the symptoms (ligula swelling, small tumors, normal tumors, heavy tumors) was significantly changed relative to SG200Δ19A-1. (**B**) The same infected plants as in (**A**) were scored for anthocyanin pigmentation. Percentage of plants displaying anthocyanin coloration is indicated. Error bars depict standard deviation. (**C**) Quantification of *tin2* gene expression during biotrophic development of *U. maydis*. Total RNA was extracted from leaves infected with SG200 at 2, 4, 6, and 8 days post infection (dpi), and also from cells grown in axenic culture in YEPSL medium. Transcript levels of *tin2* during different growth stages of *U. maydis* strain SG200 were determined by quantitative real-time PCR. Constitutively expressed *U. maydis* peptidylprolyl isomerase (*ppi*) was used for normalization. Three biological replicates were analyzed, error bars depict standard deviation. *tin2* expression in budding cells grown in axenic culture was set to 1.0. (**D**) Amino acid sequence alignment of Tin2 proteins (Um, *U. maydis*; Sr, *S. reilianum*; Us, *U. scitaminea*). Identical amino acids are boxed in yellow. Alpha helices (orange bars) and beta sheets (black arrows) are indicated. Proline-repeat sequences are marked with red lines.

The following figure supplements are available for figure 1:

**Figure supplement 1**. Substitution of the C-terminal 5 amino acids of Tin2 affects effector function.

---

*tin2* was induced during biotrophic growth (*Figure 1C*). Microscopic analysis and fungal biomass determination revealed that a SG200Δtin2 mutant did colonize plants efficiently, but was attenuated during late proliferation (*Figure 2*). To get an idea, which processes might be affected by Tin2, we determined the transcriptional response of *Z. mays* to infection by SG200 and SG200Δtin2. At 4 days post inoculation (dpi), RNA was prepared from three biological replicates and analyzed by Affymetrix maize genome arrays as described before for other *tin* mutants (*Brefort et al., in press*). Hierarchical clustering analysis of differentially regulated genes showed that the expression patterns of plant genes in response to SG200Δtin2 was related to the response to SG200 (*Figure 3A*), consistent with the small contribution of Tin2 to tumor formation. Among the top five genes most highly induced after SG200 infection (compared to SG200Δtin2), two were related to anthocyanin biosynthesis (*Figure 3—source data 1*) and one of these was the second most highly induced gene when SG200-infected tissue was compared with mock inoculated tissue (*Figure 3—source data 2*). To substantiate these data, we performed quantitative real-time PCR (qPCR) for the five genes comprising the anthocyanin pathway (*Figure 3B*) using tissue from maize infected with SG200 and SG200Δtin2, respectively (*Figure 3C*). This analysis revealed that all five genes tested were significantly induced in SG200-infected tissue, illustrating that the anthocyanin pathway is transcriptionally induced when *U. maydis* provides the *tin2* effector. Another pathway comprising three differentially regulated genes concerned the biosynthesis of the secondary metabolite 2,4-dihydroxy-7-methoxy-1,4-benzoxazin-3-one (DIMBOA, *Figure 3—source data 1*; *Figure 3D*) implicated in aphid and fungal resistance (*Ahmad et al., 2011*). When the

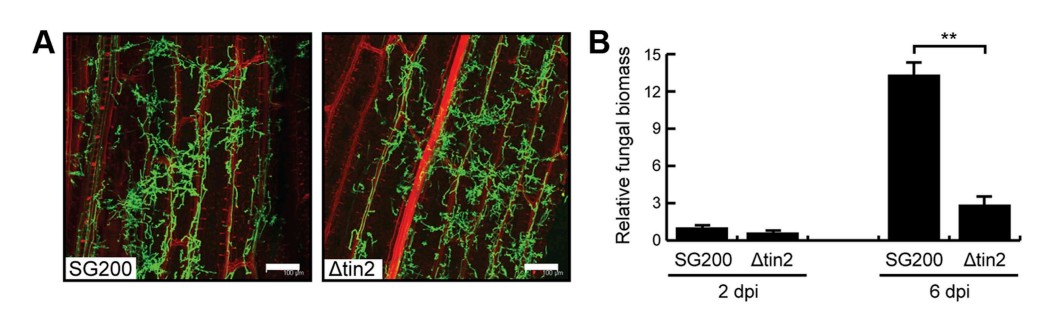

**Figure 2**. Biotrophic development of SG200 and SG200Δtin2. (**A**) Maize seedlings were infected by *U. maydis* strain SG200 and SG200Δtin2 and observed at 3 days post inoculation by confocal microscopy. Fungal hyphae were visualized by WGA-AF488 staining (green). Plant cell walls were visualized by propidium iodide staining (red). Bar = 100 μm. (**B**) Quantification of fungal biomass in SG200- and SG200Δtin2-infected tissue. Genomic DNA was extracted from leaf segments infected with SG200 and SG200Δtin2 at 2 and 6 days post inoculation. Quantitative real-time PCR was performed on genomic DNA. Relative fungal biomass was calculated from the *U. maydis ppi* gene and the *Z. mays GAPDH* gene, after setting the ratio of SG200 at 2 dpi to 1.0. Error bars indicate the standard deviation of three biological replicates. \*\*p<0.01.

expression levels of eight genes comprising the entire DIMBOA biosynthesis pathway were analyzed by qPCR, we detected significant differences in expression for *Bx1*, *Bx2*, *Bx5* and *Bx8* (**Figure 3E**) in SG200- and SG200Δtin2-infected tissue while we could not detect differences in expression for *Bx3*, *Bx4*, *Bx6* and *Bx7* (not shown). To determine whether the transcriptional changes detected can be linked to the presence or absence of respective metabolites, amounts of total anthocyanin were quantified photometrically at the anthocyanin specific absorption of 515 nm (**Figure 4A,B**). This revealed an approximately 15-fold increase of anthocyanins in leaf material upon infection with the wild-type strain that was absent in the SG200Δtin2-infected leaves (**Figure 4B**). By UPLC-PDA-TOF-MS analysis the identity of the three highly abundant anthocyanins in infected maize leaves were determined: cyanidin 3-glucoside (449.0996 Da, $C_{21}H_{21}O_{11}^+$); cyanidin 3-(6"-malonylglucoside) (535.1088 Da, $C_{24}H_{23}O_{14}^+$); cyanidin 3-(3",6"-dimalonylglucoside) (621.1092 Da, $C_{27}H_{25}O_{17}^+$) and confirmed by $MS^2$ fragmentation studies (**Figure 4—source data 1**). These cyanidin derivatives were previously described to be the major anthocyanins in maize leaves (**Fossen et al., 2001**). Relative amounts of these three cyanidins identified confirmed the overall anthocyanin profile (**Figure 4C**). In the UPLC-PDA-TOF-MS analysis, we also identified flavonoids and DIBOA-glucoside to be reduced in tissue infected with the *tin2* mutant relative to the corresponding wild type (**Figure 4D,E**). Exact mass as well as $MS^2$ spectra information allowed to identify the three flavonoids as: orientin 2"-*O*-rhamnoside (594.1585 Da, $C_{27}H_{30}O_{15}$); isoorientin 2"-*O*-rhamnoside (594.1585 Da, $C_{27}H_{30}O_{15}$); quercetin-glucoside-rhamnoside (610.1534, $C_{27}H_{30}O_{16}$) (**Figure 4—source data 1**). The biosynthesis of these flavonoids branches off the anthocyanin pathway and is likely a side effect of the upregulation of this pathway after infection by wild-type *U. maydis* (**Figure 3B**). In case of DIMBOA biosynthesis we identified 2,4-dihydroxy-1,4-benzoxazin-3-one glucoside (DIBOA-glucoside, 343.0903 Da, $C_{14}H_{17}NO_9$, **Figure 4E**) to be regulated by Tin2. These findings suggest that, besides the influence of Tin2 on anthocyanin, additional pathways may be affected directly or indirectly by *tin2*.

## Functional domains in Tin2

The Tin2 protein does not reveal any conserved InterPro domains. Orthologous proteins are absent in other organisms examined to date but exist in the related smut fungi *Sporisorium reilianum* and *U. scitaminea*. An amino acid alignment revealed significant sequence conservation in the C-terminal third of the protein, whereas the N-terminus showed less conservation (**Figure 1D**). The C-terminal region contains two proline repeat sequences (PxxP), suggesting that this domain might be important for the interaction with other proteins (**Saksela et al., 1995**; **Kay et al., 2000**; **Dornan et al., 2003**) (**Figure 1D**). Based on the high conservation of the C-terminus (**Figure 1D**) a series of C-terminally truncated Tin2 proteins were expressed in SG200Δ19A-1 (**Brefort et al., in press**) to test for complementation of tumor formation and anthocyanin induction (**Figure 1A,B**). Tin2$_{1-206}$ truncated by one amino

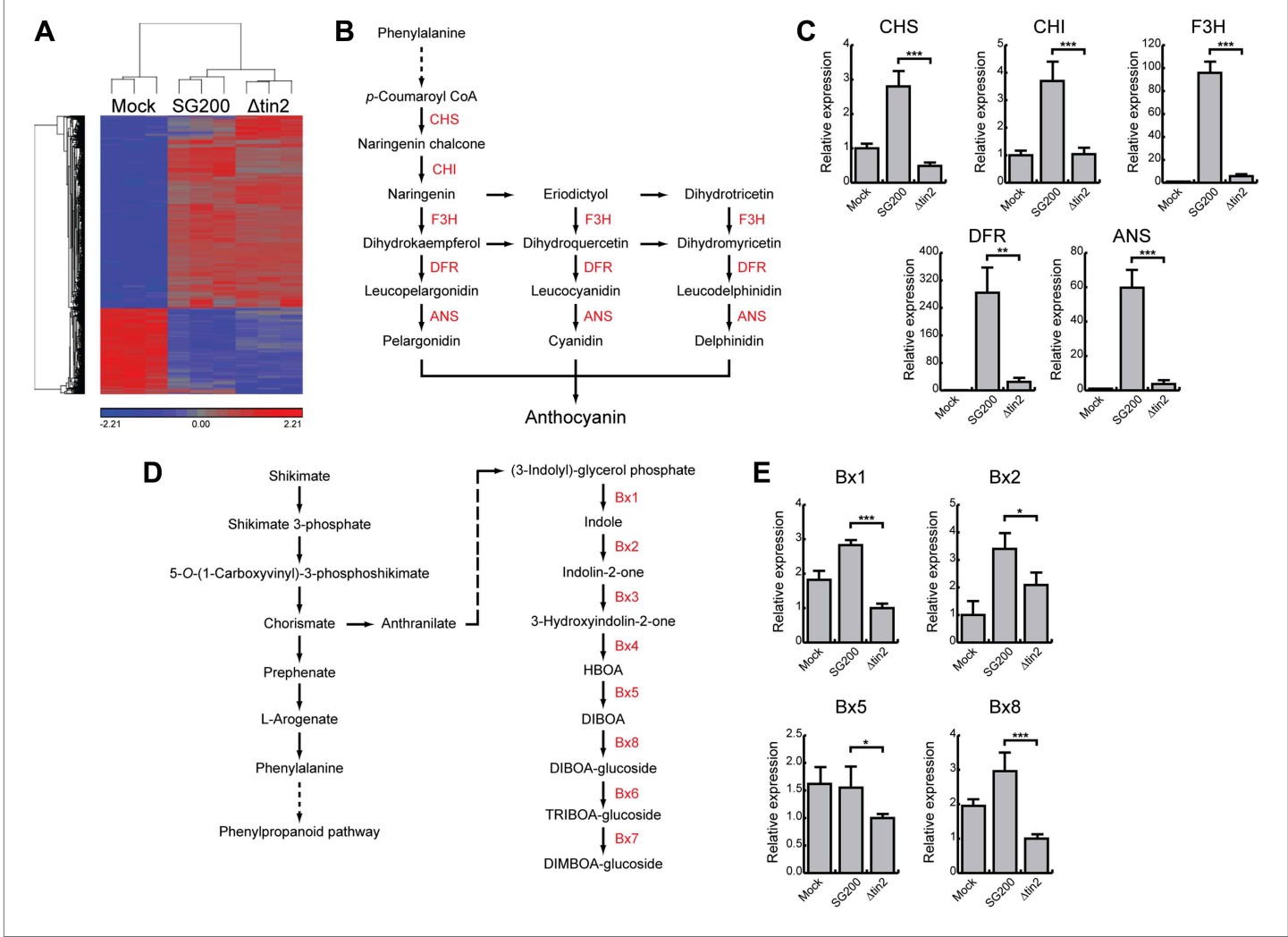

**Figure 3**. Differentially expressed maize genes in leaves infected with SG200 and SG200Δtin2. (**A**) Plant transcriptome analysis of SG200- and SG200Δtin2-infected tissue. Hierarchical clustering was performed by the Partek Genomics Suite version 6.12 to visualize expression of maize genes transcriptionally regulated 4 days after mock inoculation (left) infection by *U. maydis* strain SG200 (middle) and infection by SG200Δtin2 (right). The X-axis depicts clustering of the microarray samples for each of the three biological replicates per inoculation. The Y-axis shows clustering of the regulated maize transcripts based on similarity of their expression patterns. Red: upregulated genes; blue: downregulated genes. (**B**) Schematic model of the flavonoid biosynthetic pathway in maize. The enzymes involved in the discrete biosynthetic steps are shown in red. CHS, chalcone synthase; CHI, chalcone isomerase; F3H, flavanone 3-hydroxylase; DFR, dihydroflavonol 4-reductase; ANS, anthocyanin synthase. (**C**) qPCR based quantification of the expression levels of genes from the flavonoid pathway after infection with indicated *U. maydis* strains and collecting infected leaf material 6 dpi. **$p<0.01$, ***$p<0.001$. (**D**) Schematic model of the DIMBOA biosynthetic pathway in maize. The enzymes involved in the discrete biosynthetic steps are shown in red. Bx1, indole-3-glycerol phosphate lyase; Bx2, indole monooxygenase; Bx3, indolin-2-one monooxygenase; Bx4, 3-hydroxyindolin-2-one monooxygenase; Bx5, HBOA monooxygenase; Bx8, DIBOA UDP-glucosyltransferase; Bx6, 2-oxoglutarate-dependent dioxygenase; Bx7, TRIBOA-glucoside methyltransferase. (**E**) qPCR based quantification of the expression levels of genes from the DIMBOA pathway after infection with indicated *U. maydis* strains and collecting infected leaf material 6 dpi. *$p<0.05$, ***$p<0.001$.

The following source data are available for figure 3:

**Source data 1**. List of upregulated maize genes after SG200 infection (vs SG200Δtin2).

**Source data 2**. List of differentially regulated maize genes after SG200 infection (vs mock).

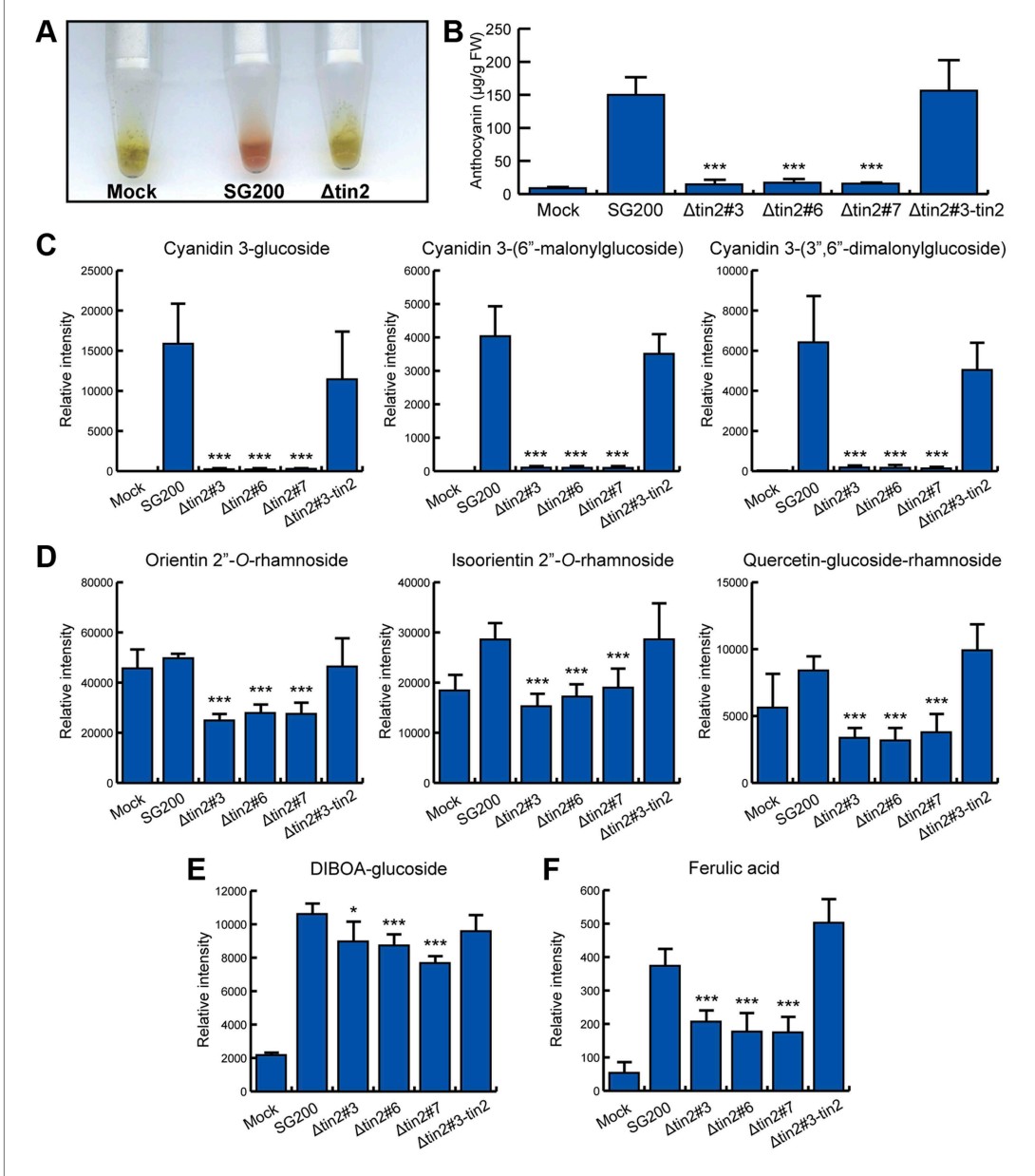

Figure 4. Identification of anthocyanins and other metabolites in infected maize leaves. (**A**) Anthocyanin was extracted from leaves syringe-inoculated with H₂O (mock), SG200, and SG200Δtin2#3. Two cm long leaf segments located 1 cm below the injection holes were excised, frozen in liquid nitrogen, ground and extracted with 100% ethanol. For each sample, 5 leaf segments were combined. The supernatant from SG200-infected leaves showed red color after acidification with concentrated HCl while the other two samples stayed green. (**B–E**) Infected leaf segments from mock-inoculated leaves and leaves infected with SG200, three independent SG200Δtin2 mutants and SG200Δtin2#3-tin2 were collected at 6 dpi. Excised leaf segments were ground in liquid nitrogen and served as a starting material for all subsequent analyses. (**B**) Measurement of total anthocyanin content. Anthocyanins were extracted and their amounts were measured at 515 nm using cyanidin 3-arabinoside chloride for calibration. Error bars indicate the standard deviation of three biological replicates. ***$p < 0.001$ (vs SG200). (**C**) Specific accumulation of the three major anthocyanins: cyanidin 3-glucoside, cyanidin 3-(6"-malonylglucoside), and cyanidin 3-(3",6"-dimalonylglucoside). Powdered samples were extracted and the polar phase was measured by UPLC-PDA-TOF-MS as described (***Djamei et al., 2011***). Data shown are representative of three biological replicates. ***$p < 0.001$ (vs SG200). (**D**) Decreased accumulation of the three flavonoids in SG200Δtin2-infected tissue: orientin 2"-*O*-rhamnoside, isoorientin 2"-*O*-rhamnoside, and quercetin-glucoside-rhamnoside. Powdered samples were extracted and the polar phase was measured by UPLC-PDA-TOF-MS as described (***Djamei et al., 2011***). ***$p < 0.001$ (vs SG200). (**E**) Decreased accumulation of DIBOA-glucoside in SG200Δtin2-infected tissue. Powdered samples were extracted and the polar phase was measured by UPLC-PDA-TOF-MS as described (***Djamei et al., 2011***). Data shown are representative of three biological replicates. *$p < 0.05$, ***$p < 0.001$ (vs SG200). (**F**) Decreased

*Figure 4. Continued on next page*

*Figure 4. Continued*

accumulation of ferulic acid in SG200Δtin2-infected tissue: powdered samples were extracted and the polar phase was measured by UPLC-PDA-TOF-MS as described (***Djamei et al., 2011***). Data shown are representative of three biological replicates. \*\*\*p<0.001 (vs SG200).
The following source data are available for figure 4:

**Source data 1**. Markers identified by metabolite fingerprinting (UPLC-ESI-TOF-MS analysis) in leaves of *Z. mays* 6 days post infection and verified by UHPLC-ESI-QTOF-MS/MS analysis or coelution.

acid complemented as efficiently as full-length Tin2$_{1-207}$ (***Figure 1A,B***), whereas Tin2$_{1-202}$ lacking five amino acid residues at the C-terminus could neither complement for tumor formation nor anthocyanin induction (***Figure 1A,B***). Tin2$_{AAAAA}$ protein, in which the five most C-terminal amino acids were substituted by alanine, was also largely unable to complement (***Figure 1—figure supplement 1***). These results indicated that the C-terminal region of Tin2 is crucial for tumor formation as well as anthocyanin induction.

## Tin2 functions within the plant cytosol

To demonstrate that the Tin2 protein can be secreted, we constructed a strain constitutively expressing a biologically active Tin2-HA fusion protein (***Figure 5A,B***) and showed that full-length Tin2-HA was detectable in the culture supernatant (***Figure 5C***). To visualize protein secretion during biotrophic development a Tin2$_{1-25}$-mCherry-Tin2$_{26-207}$ fusion protein was expressed in SG200Δtin2 under the *cmu1* promoter, which confers strong expression during plant colonization (***Djamei et al., 2011***). After infection, mCherry fluorescence could be observed around fungal hyphae (***Figure 6A***). When subjected to plasmolysis, which enlarges the apoplastic compartment, the fluorescence of Tin2$_{1-25}$-mCherry-Tin2$_{26-207}$ resided in the apoplast (***Figure 6B***), demonstrating secretion (***Doehlemann et al., 2009***). Strain SG200Δtin2-Tin2$_{1-25}$-mCherry-Tin2$_{26-207}$ elicited anthocyanin induction, but compared to SG200 the appearance was delayed (***Figure 6—figure supplement 1***), suggesting that the Tin2$_{1-25}$-mCherry-Tin2$_{26-207}$ fusion protein is only partially functional. In addition, mCherry-Tin2$_{26-207}$ was never detected in the cytosol of infected plants. Since anthocyanin biosynthesis is occurring in the cytosol of plant cells (***Holton and Cornish, 1995***), we expected that Tin2 would translocate to plant cells after being secreted. To demonstrate a function of Tin2 in the plant cytosol, we expressed *tin2* without its secretion signal under the CaMV 35S promoter in maize leaf epidermal cells after biolistic bombardment. As this

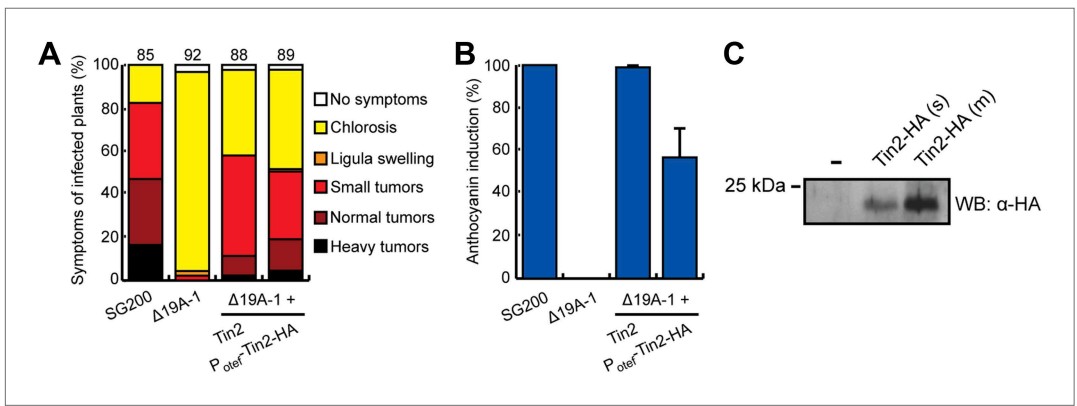

**Figure 5**. Biological activity of Tin2-HA and demonstration of its secretion. SG200Δ19A-1-P$_{otef}$Tin2-HA expresses Tin2-HA protein under the constitutive *otef* promoter (***Spellig et al., 1996***). (**A**) Biological activity of Tin2-HA protein was confirmed by assaying virulence of SG200Δ19A-1-P$_{otef}$Tin2-HA and comparing this to SG200Δ19A-1-Tin2. (**B**) Complementation of anthocyanin induction by SG200Δ19A-1 strains by expressing Tin2-HA protein. The same infected plants as in (**A**) were scored for anthocyanin induction, which is given in % of all plants infected. (**C**) Detection of Tin2-HA protein in culture supernatants. Cells of SG200Δ19A-1 (−) and SG200Δ19A-1-P$_{otef}$Tin2-HA (s, single integration; m, multiple integration) were grown in CM liquid medium to an OD$_{600}$ of 0.5. Proteins in supernatants were collected after TCA/DOC-precipitation and used in western blot analysis. The western blot was developed with anti-HA antibody (Sigma-Aldrich).

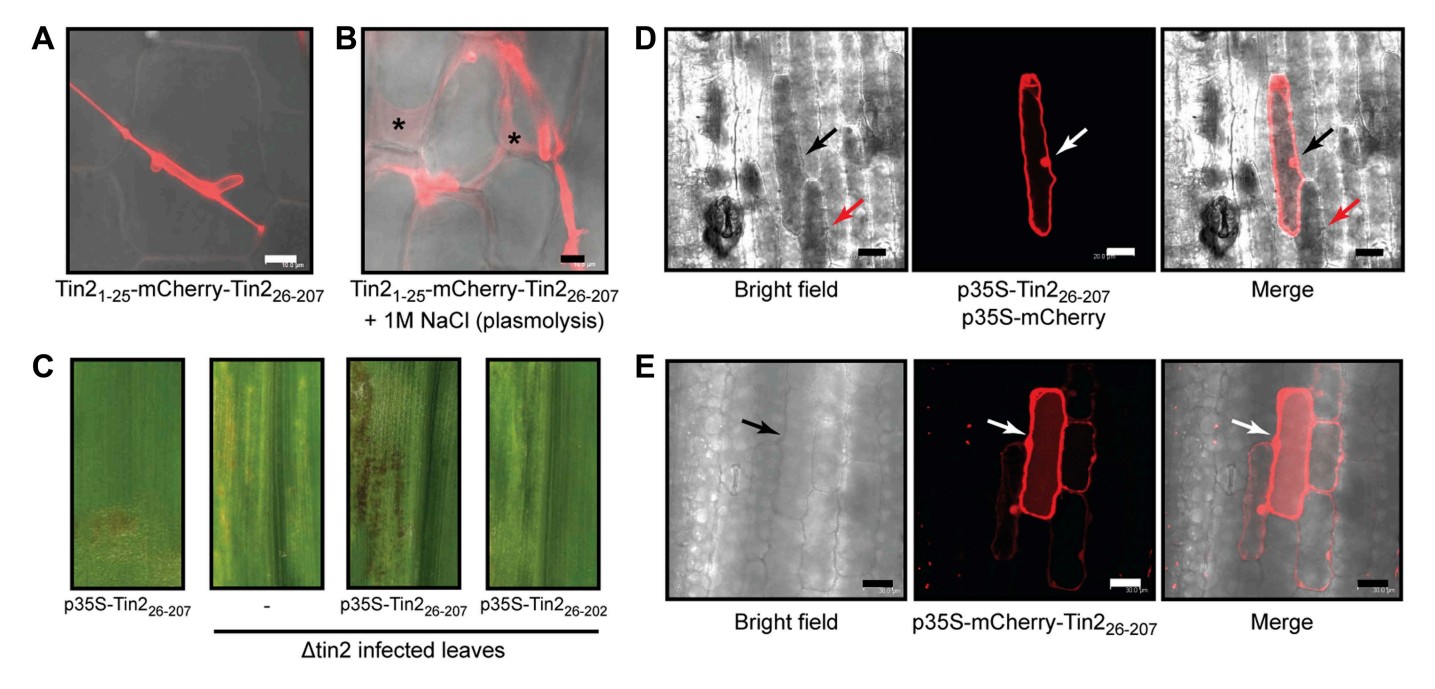

**Figure 6**. Tin2 protein secretion during biotrophic development and its function inside plant cells. (**A**) Tin2 protein secretion during plant infection. Leaves are infected with SG200Δtin2-Tin2$_{1-25}$-mCherry-Tin2$_{26-207}$. mCherry fluorescence is observed at 3 dpi. Bar = 10 µm. (**B**) Same as (**A**) but treated with 1 M NaCl to induce plasmolysis. Asterisks mark enlarged apoplastic spaces. Bar = 10 µm. (**C**) Anthocyanin induction by transient expression of Tin2$_{26-207}$. Plasmid constructs indicated were delivered by biolistic gene transfer in maize leaves or leaves infected with SG200Δtin2. (**D**) Confocal microscopy of cells co-expressing Tin2$_{26-207}$ and mCherry. Dark staining indicates anthocyanin accumulation. Bar = 20 µm. (**E**) Confocal microscopy of mCherry-Tin2$_{26-207}$ transiently expressed in maize epidermal cells. The initially transformed cell is indicated by an arrow. Bar = 30 µm.

The following figure supplements are available for figure 6:

**Figure supplement 1**. Biological activity of secreted mCherry-Tin2$_{26-207}$.

did not induce anthocyanin (*Figure 6C*, leftmost panel), we repeated the experiment with leaves from plants infected with SG200Δtin2. Here, anthocyanin could be induced by Tin2$_{26-207}$ (*Figure 6C*, third panel), was absent in infected leaves without *tin2* expression (*Figure 6C*, second panel) and could also not be observed when biologically inactive Tin2$_{26-202}$ was expressed (*Figure 6C*, rightmost panel). To confirm that anthocyanin is produced in SG200Δtin2-infected tissue transiently transformed with *tin2*, p35S-Tin2$_{26-207}$ and p35S-mCherry as transformation control were introduced simultaneously in SG200Δtin2-infected tissue. Anthocyanin production, detected by its dark color in bright field microscopy, was observed in the bombarded cell that also expressed cytosolic mCherry (*Figure 6D*). Interestingly, cells neighboring the transformed cell also accumulated anthocyanin (*Figure 6D*, red arrow). When biolistic transformation of uninfected maize leaves was carried out with a plasmid expressing mCherry-Tin2$_{26-207}$, the fusion protein was found to spread into non-transformed neighboring cells (*Figure 6E*). Symplastic spreading was observed for the Cmu1 effector of *U. maydis* and for several translocated effectors of *Magnaporthe oryzae* (*Khang et al., 2010*; *Djamei et al., 2011*). Collectively, these results demonstrate that Tin2 functions within the plant cytosol and suggest that Tin2 may prepare maize cells for the infection by *U. maydis*.

## Tin2 stabilizes a maize protein kinase

To gain insight into the plant processes affected by Tin2, we identified Tin2 interacting proteins by yeast two-hybrid screening. After rescreening, only one putative cytoplasmic serine/threonine protein kinase from maize, designated ZmTTK1 (Tin2-targeting kinase 1) could be confirmed (not shown). ZmTTK1 is an uncharacterized kinase, with its kinase domain located at the C-terminus (*Figure 7A*). Maize may contain two putative paralogs of ZmTTK1 (GRMZM2G088409; GRMZM2G068192) with highly conserved kinase domains (63 and 65% identity), but reduced homology in the N-terminal

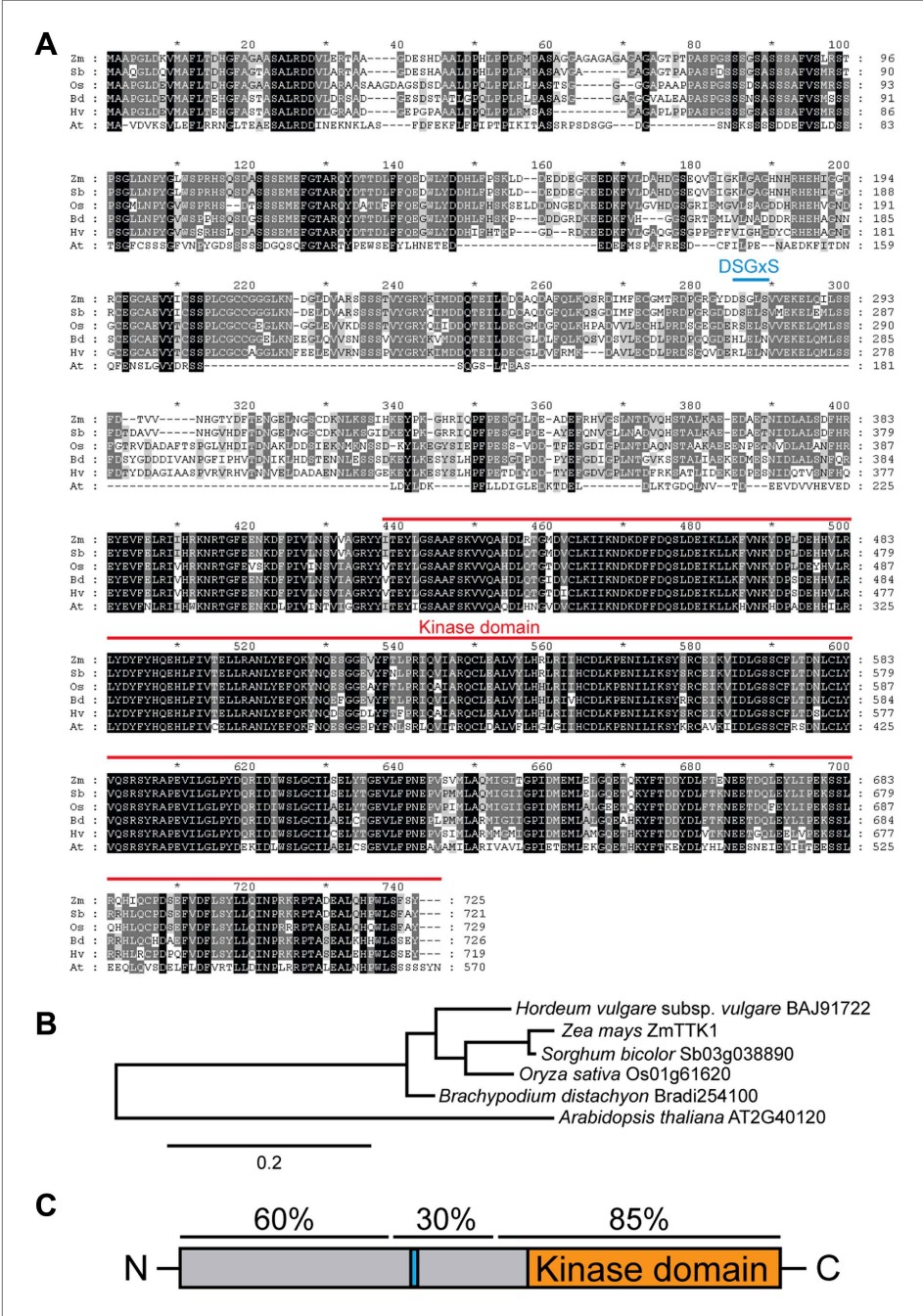

**Figure 7**. Comparison of ZmTTK1 orthologs from different grasses. (**A**) Amino acid sequence alignment of ZmTTK1 orthologs. ZmTTK1 orthologs (Zm, *Zea mays*) could be identified in *Sorghum bicolor* (Sb), *Oryza sativa* (Os), *Hordeum vulgare* (Hv), *Brachypodium distachyon* (Bd) and a related protein found in *Arabidopsis thaliana* (At). Amino acids conserved in all 5 orthologs from monocot plants and the related protein in *A. thaliana* are highlighted in black, amino acids conserved in 4 or 5 of 6 these proteins are highlighted in dark gray and amino acids conserved in 3 of the 6 proteins are highlighted in light gray. The phosphodegron-like motif is indicated by a blue line. The kinase domain is indicated by a red line. (**B**) Phylogenetic tree of ZmTTK1 orthologs in monocot plants and the related protein from *A. thaliana*. Amino acid sequences of these proteins were analyzed by Phylogeny.fr to calculate the phylogenetic relationship. Bar indicates evolutionary distance. Accession numbers are indicated. (**C**) Schematic structure of ZmTTK1. The N-terminal region is labeled in gray and the C-terminal kinase domain labeled in orange. The phosphodegron-like motif is indicated in blue. Amino acid sequence identity of the N-terminal middle and C-terminal regions among orthologs in monocots is indicated above.

domains (34% and 36% identity, respectively). Orthologs are found in monocot plants like *Sorghum bicolor*, *Oryza sativa*, *Hordeum vulgare*, and *Brachypodium distachyon* and these display high amino acid sequence identity with ZmTTK1 from maize (between 75% and 94%) (*Figure 7A,B*). *Arabidopsis thaliana* also encodes a putative ortholog of ZmTTK1 (AT2G40120), but in this case the amino acid identity is significantly lower (57%) (*Figure 7A,B*). ZmTTK1 orthologs from monocot plants display highly conserved N- and C-terminal domains, which are separated by a more variable central domain (*Figure 7A,C*). In the related *A. thaliana* protein, the N-terminal region shows only patchy homology to the monocot proteins and the central domain is missing (*Figure 7A*). None of these proteins have been functionally characterized to date.

In yeast two-hybrid assays, full-length ZmTTK1 was able to interact with $Tin2_{26-207}$ and $Tin2_{26-206}$, while no interaction was observed with $Tin2_{26-202}$ lacking biological activity (*Figure 8A*). Physical interaction between recombinant Strep-ZmTTK1 protein and recombinant $GST-Tin2_{26-207}$ protein was demonstrated by co-IP using $GST-Tin2_{26-202}$ as negative control (*Figure 9A*). Recombinant ZmTTK1 purified from *E. coli* was expressed as full-length protein (*Figure 9B*) and showed in vitro kinase activity determined with MBP (myelin basic protein) as substrate. However, no significant differences in MBP phosphorylation were apparent in the presence or absence of Tin2 (*Figure 9C*). To examine whether Tin2 protein affects ZmTTK1 activity in vivo, we transiently expressed ZmTTK1 protein in *Nicotiana benthamiana*. In this system, ZmTTK1 protein proved highly unstable and the N-terminal region could not be detected (*Figure 10A*). This region contains a phosphodegron-like motif DSGxS, which, when phosphorylated at serine residues, serves as a recognition motif for the ubiquitin ligase complex leading to protein degradation via the ubiquitin-proteasome system (*Ravid and Hochstrasser, 2008*; *Spoel et al., 2009*). Addition of proteasome inhibitor MG132 to *N. benthamiana* cells transiently expressing ZmTTK1-HA allowed detection of full-length protein (*Figure 10B*). In addition, $ZmTTK1_{S279/282A}$-HA protein carrying alanine substitutions at positions serine 279 and serine 282 in the phosphodegron-like motif could be expressed as full-length protein in the transient *N. benthamiana* system (*Figure 10B*). Furthermore, poly-ubiquitinated ZmTTK1-HA protein accumulated in ZmTTK1-HA expressing plant cells treated with MG132 but not in $ZmTTK1_{S279/282A}$-HA expressing *N. benthamiana* cells (*Figure 10C*). These results indicate that the phosphodegron-like motif in ZmTTK1 promotes partial degradation via the ubiquitin-proteasome system.

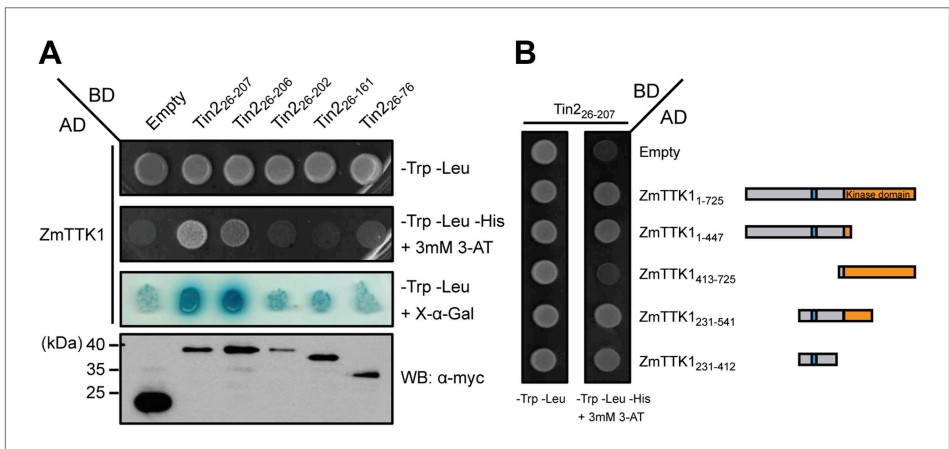

**Figure 8**. Mapping the interaction regions between Tin2 and ZmTTK1 by yeast two-hybrid analysis. (**A**) A series of C-terminally truncated Tin2 proteins were expressed in fusion with GAL4BD (BD) protein and their interaction with full-length ZmTTK1 protein fused with GAL4AD (AD) were tested by yeast two-hybrid assay. Yeast transformants were grown on SD medium lacking indicated amino acids. Interaction was assessed from growth on SD -Trp -Leu -His + 3 mM 3-AT medium and from SD -Trp -Leu + X-α-Gal medium. Protein expression was verified by western blot with anti-myc antibody (Sigma-Aldrich) (bottom panel). (**B**) Mapping of the Tin2-interacting region of ZmTTK1. A series of truncated ZmTTK1 genes were fused with GAL4AD (AD) protein and their interaction with $Tin2_{26-207}$ protein fused with GAL4BD (BD) was assessed by yeast two-hybrid assay. Interaction was shown by growth of respective strains on SD -Trp -Leu -His + 3 mM 3-AT medium. Truncated ZmTTK1 versions are drawn schematically, the kinase domain is depicted in orange, and the N-terminal domain is colored in gray. The phosphodegron-like motif DSGxS is indicated in blue.

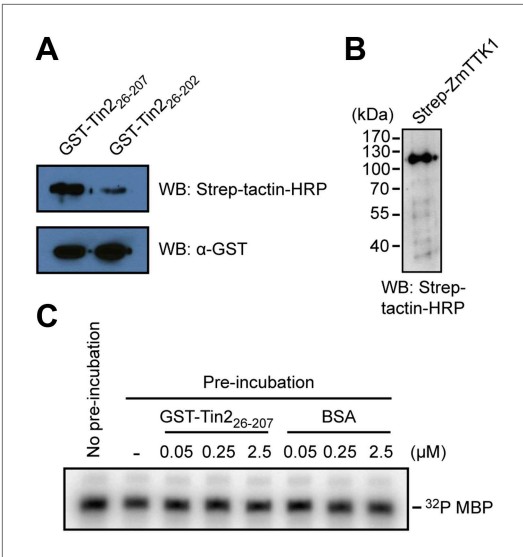

**Figure 9**. Physical interaction of Tin2 protein with full-length ZmTTK1 and in-vitro kinase activity of ZmTTK1. (**A**) Physical interaction of Tin2 and ZmTTK1 demonstrated by in vitro GST-pull down assay using recombinant proteins. Recombinant GST-Tin2$_{26-207}$ or GST-Tin2$_{26-202}$ protein bound to glutathione sepharose beads (GE healthcare), respectively, was incubated with extract from induced BL21 (DE3)/pPRIBA102-ZmTTK1 at 4°C for 1 hr. GST fusion proteins were eluted with reduced glutathione. Strep-ZmTTK1 in eluate was detected by western blot (WB) using Strep-tactin-HRP and α-GST antibody for detection, respectively. Top panel detects precipitated Strep-ZmTTK1, bottom panel detects input GST-Tin2 fusion proteins. (**B**) Full-length recombinant Strep-ZmTTK1 expressed and purified from *E. coli* was detected by western blot analysis with Strep-tactin-HRP. (**C**) In vitro kinase activity of recombinant Strep-ZmTTK1 protein. Recombinant GST-Tin2$_{26-207}$ and Strep-ZmTTK1 proteins at indicated concentration were pre-incubated for 60 min at 4°C followed by addition of γ-$^{32}$P ATP and myelin basic protein (MBP). Bovine serum albumin (BSA) was used as a control. Incubation continued at 28°C for 30 min. Proteins were separated by SDS-PAGE and phosphorylated protein was detected.

The region in ZmTTK1 that interacts with Tin2 was mapped by yeast two-hybrid assays and shown to coincide with the variable region containing the phosphodegron-motif (*Figure 8B*). Next, when ZmTTK1-HA was co-expressed in *N. benthamiana* together with functional Tin2$_{26-207}$, we detected full-length ZmTTK1, but not when non-functional truncated Tin2$_{26-202}$ was co-expressed with ZmTTK1 (*Figure 10D,E*). Furthermore, immunoprecipitated ZmTTK1-HA co-expressed with Tin2$_{26-207}$ showed MBP phosphorylation activity in vitro while expression of ZmTTK1-HA alone or co-expression of ZmTTK1-HA with truncated Tin2$_{26-202}$ did not yield an immunoprecipitate with phosphorylation activity (*Figure 10E*, bottom panel). This illustrates that Tin2 protein targets the phosphodegron-containing region of ZmTTK1, stabilizing full-length, active ZmTTK1.

To rule out that these effects are specific to the heterologous *N. benthamiana* system, ZmTTK1-YFP was transiently expressed in maize with and without co-expression of mCherry-Tin2$_{26-207}$. When ZmTTK1-YFP and mCherry-Tin2$_{26-207}$ were co-expressed, the number of cells showing a YFP signal was increased significantly compared to the situation when ZmTTK1-YFP was expressed alone, and in addition, a significant percentage of cells showed strong YFP fluorescence in the co-expression conditions (*Figure 11A,B*). This suggests that the ZmTTK1-YFP fusion protein is also unstable in maize epidermal cells. Co-expression of Tin2 stabilizes the fusion protein, similar to what was observed in *N. benthamiana*.

To elucidate whether there is a direct link between ZmTTK1 activity and anthocyanin induction, the stabilized form of the kinase, ZmTTK1$_{S279/282A}$, was transiently co-expressed with cytoplasmic mCherry in leaves infected with SG200Δtin2. In transiently transformed cells weak anthocyanin induction could be observed (*Figure 11C*). In maize anthocyanin biosynthesis genes are positively regulated via ZmR1 and ZmC1 transcription factors (*Ludwig and Wessler, 1990*;

*Dooner et al., 1991*). Therefore, we co-expressed Tin2$_{26-207}$ and ZmTTK1 in uninfected maize leaves together with mCherry-ZmR1 or mCherry-ZmC1 protein, respectively. mCherry-ZmC1 localized to the nucleus, and this localization did not change significantly in the presence of Tin2$_{26-207}$ and ZmTTK1 (*Figure 11D*). When mCherry-ZmR1 was expressed alone, the fusion protein showed weak cytoplasmic and nuclear localization (*Figure 11E*). However, when co-expressed with Tin2$_{26-207}$ and ZmTTK1, mCherry-ZmR1 protein showed almost exclusively intense nuclear localization (*Figure 11D,E*). These results suggest that ZmTTK1 protein kinase stabilized by Tin2 might phosphorylate the ZmR1 protein, which is then imported into the nucleus (or is retained there) and induces anthocyanin biosynthetic genes.

## Tin2 differentially affects anthocyanin and lignin biosynthesis genes

Since anthocyanin precursors accumulate in the plant cytosol and anthocyanin is transported to the vacuole, two compartments *U. maydis* is not in direct contact with due to its biotrophic life style,

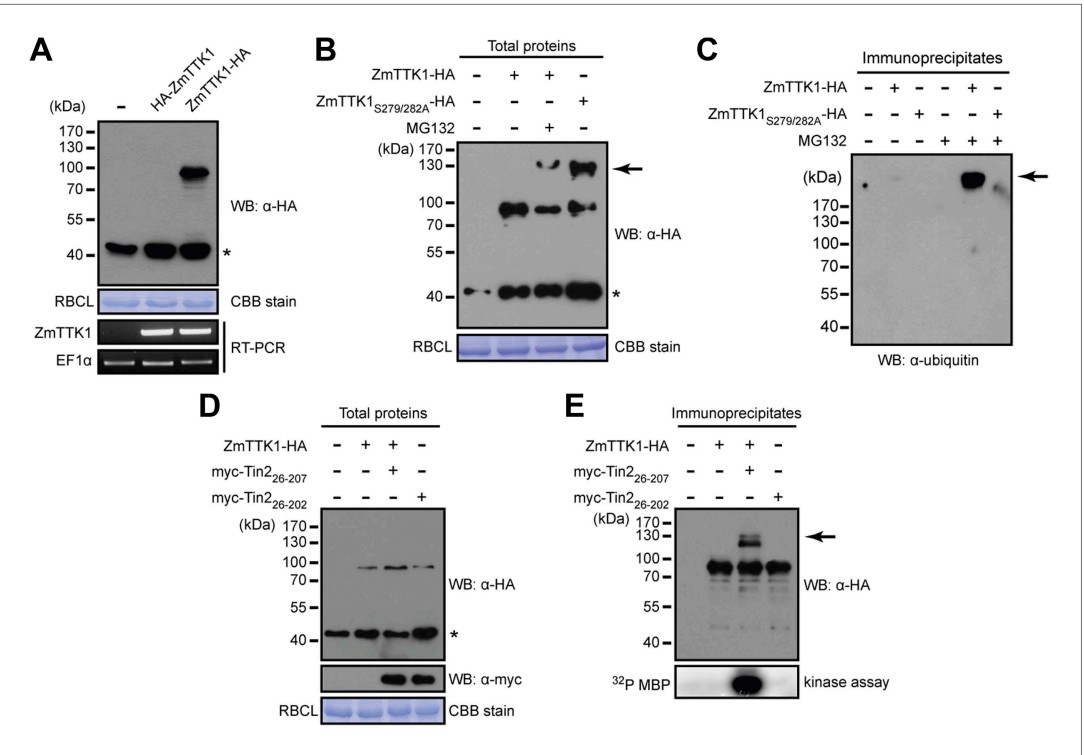

**Figure 10**. Tin2 protein stabilizes cytoplasmic maize protein kinase ZmTTK1. (**A**) ZmTTK1-HA and HA-ZmTTK1 expression in *N. benthamiana*. HA-ZmTTK1 or ZmTTK1-HA protein was transiently expressed in *N. benthamiana* after infiltration with the respective *A. tumefaciens* strains GV3101 carrying pBIN19AN-HA-ZmTTK1 and pBIN19AN-ZmTTK1-HA. These plasmids express the indicated fusion proteins with a C-terminal IgG binding site. Expression was shown by western blot (WB) using indicated antibody. Asterisk labels a non-specific band. Rubisco large subunit (RBCL) stained with coomassie brilliant blue (CBB) served as a loading control. ZmTTK1-HA transcripts were analyzed by RT-PCR, EF1α served as a control. (**B**) Protein expression of ZmTTK1-HA with proteasome inhibitor MG132 (100 μM) and ZmTTK1$_{S279/282A}$-HA. (**C**) Detection of poly-ubiquitinated ZmTTK1. After immunoprecipitation with human IgG-agarose, proteins were subjected to western blot to detect poly-ubiquitinated ZmTTK1 protein (arrow). The western blot was developed with monoclonal anti-ubiquitin antibody (Sigma-Aldrich). (**D**) Co-expression of ZmTTK1-HA with myc-Tin2. Total protein was analyzed by western blot using indicated antibodies. (**E**) Immunoprecipitated ZmTTK1-HA protein from (**D**) was analyzed by western blot. Kinase activity of immunoprecipitated samples shown on top was analyzed using MBP as a substrate (bottom).

we next considered that induction of the anthocyanin pathway might deplete precursors for other pathways, which could be harmful for *U. maydis* development. In the general phenylpropanoid pathway, phenylalanine is converted to 4-coumaroyl CoA, which then serves as a precursor for both, anthocyanins and lignin (*Figure 12A*). Since we had shown that all genes in the anthocyanin branch of the pathway are upregulated in tissue infected with SG200, but not in tissue infected with SG200Δtin2 (*Figure 3*), we determined abundance of transcripts encoding key enzymes in the general and lignin-specific branches of the phenylpropanoid pathway by qPCR in leaves infected with strains SG200 and SG200Δtin2. qPCR analysis demonstrated that the phenylalanine ammonia lyase (*PAL*) gene in the general part of the pathway was significantly upregulated in both SG200- and SG200Δtin2-infected tissue compared to mock inoculation (*Figure 12B*). One of the two 'cinnamoyl-CoA reductase' (*CCR*) genes of maize one was not induced, while the other one (GRMZM2G131836) was clearly induced after *U. maydis* infection, but there was no difference in induction in SG200- and SG200Δtin2-infected leaves (*Figure 12*). The major 4-coumarate-CoA ligase (*4CL*) gene in the lignin-specific branch (GRMZM2G075333; based on homology with sorghum; *Saballos et al., 2012*) was not differentially regulated, while the flavonoid-related *4CL* (GRMZM2G054013) was downregulated in SG200Δtin2-infected tissue compared to SG200-infected tissue (*Figure 12B*). Of the six cinnamyl alcohol dehydrogenase (*CAD*) genes, four genes were significantly upregulated in SG200Δtin2-infected tissue (*Figure 12B*). The enhanced expression of several

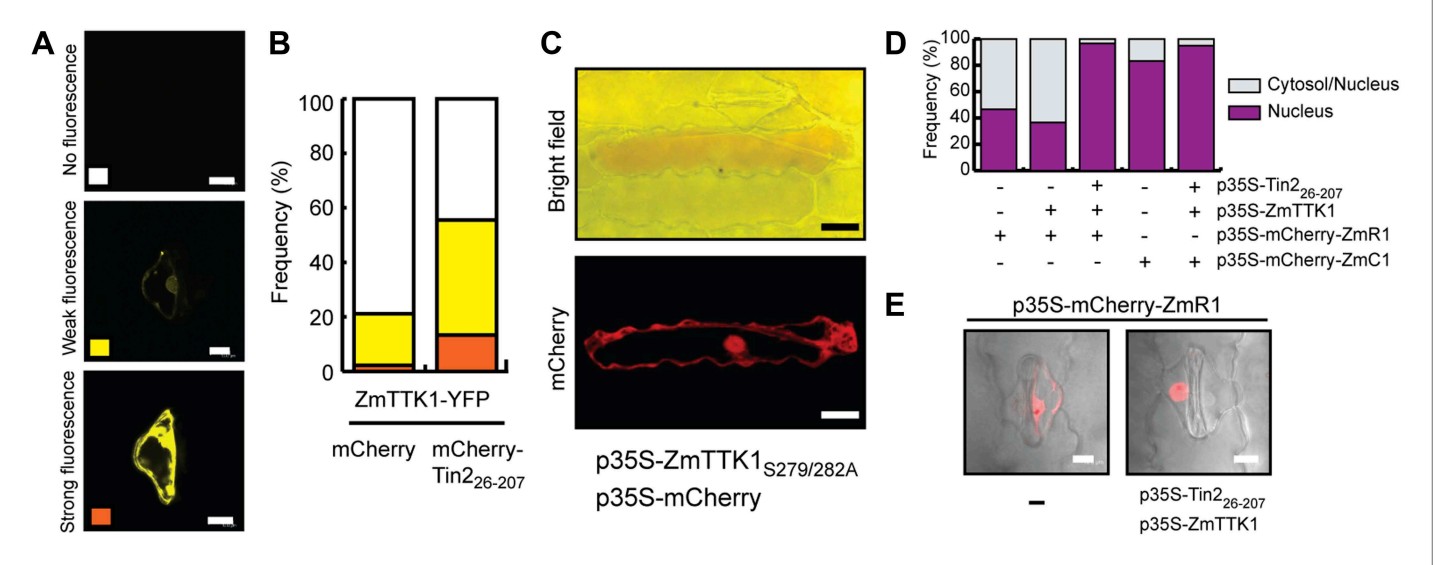

**Figure 11**. ZmTTK1 effects on localization of maize transcription factors and anthocyanin biosynthesis. (**A**) Fluorescence intensity of ZmTTK1-YFP co-expressed with mCherry-Tin2$_{26-207}$ or with mCherry, respectively, in transiently transformed maize cells. Transformed maize cells were first identified by their mCherry fluorescence (not shown) and these cells were then screened for YFP fluorescence. The three examples depict the range of ZmTTK1-YFP expression patterns seen. (**B**) Quantification of the three YFP fluorescence patterns indicated in (**A**). Expressed proteins are listed below panels. In total, 90 transformed cells were scored for YFP fluorescence from three independent experiments. A statistically significant increase of weak and strong fluorescence patterns is observed when ZmTTK1-YFP is co-expressed with mCherry-Tin2$_{26-207}$ compared to co-expression of ZmTTK1-YFP with mCherry. (**C**) Anthocyanin induction after transient expression in maize. p35S-ZmTTK1$_{S279/282A}$ and p35S-mCherry were co-expressed in SG200Δtin2-infected leaves. Anthocyanin was visualized by bright field microscopy 3 days after particle bombardment. Bar = 20 μm. (**D**) mCherry-ZmR1 or mCherry-ZmC1 protein localization pattern. mCherry fluorescence in 60 cells from three different leaves was classified into cytoplasmic and nuclear localization or exclusive nuclear localization. (**E**) Confocal microscopy of mCherry-ZmR1. p35S-mCherry-ZmR1 was transiently introduced into maize either alone (−) or together with p35S-Tin2$_{26-207}$ and p35S-ZmTTK1. mCherry-ZmR1 was visualized. Bar = 10 μm.

lignin biosynthesis genes after infection with the *tin2* mutant coincided with an transcriptional upregulation of several plant cell wall remodeling genes (*Figure 12—source data 1*) and a decrease in ferulic acid (*Figure 4F*), which is involved in crosslinking hemicellulosic polysaccharides and provides a nucleation site for lignification (*Grabber et al., 2002*), suggesting a connection between these processes. Furthermore, in a maize chalcone synthase mutant (*c2*), in which the biosynthetic pathway to anthocyanins is blocked, and in which there could conceivably be more precursor going into the lignin branch, virulence of *U. maydis* SG200 was attenuated (*Figure 13*).

Quantification of total lignin content from isolated cell walls in wild-type maize (cv. Early Golden Bantam) revealed a small increase in leaf tissue infected with *tin2* mutant strains relative to SG200-infected material that was statistically significant in one of three analyzed mutants (*Figure 14A*). Collectively, these results suggest that anthocyanin induction by Tin2 protein may negatively affect the lignin pathway. To determine whether lignin content affects virulence of *U. maydis*, we infected single and double *brown midrib* (*bm*) mutants of maize with SG200 (*Figure 14B–D*). The *bm1* mutation reduces CAD activity (*Halpin et al., 1998*) as a result of a mutation in the *ZmCAD2* gene (*Chen et al., 2012*), the major CAD involved in lignification of vascular tissue in leaves and stems. The leaf midribs of the *bm2* mutant contain less lignin, and the tissue-specific pattern of lignification in the leaves is altered (*Vermerris and Boon, 2001*). Lignin content in the leaves and stems of the *bm3* mutant is reduced, lignin subunit composition has shifted towards a higher guaiacyl/syringyl (G/S) ratio, and novel benzodioxane structures are present as a result of the incorporation of 5-hydroxyconiferyl alcohol (*Lapierre et al., 1988*; *Chabbert et al., 1994*; *Marita et al., 2003*). This is the result of a mutation in the gene encoding caffeic acid *O*-methyl transferase (*COMT*) (*Vignols et al., 1995*). The *bm4* mutation has not yet been mapped to a specific gene, but affects the cell wall composition of leaves and lowers the lignin content of leaf midribs (*Vermerris et al., 2010*). Except for the *bm2* mutant, all other single *bm* mutants displayed elevated virulence symptoms after *U. maydis* infection, with *bm3* showing the strongest effect (*Figure 14C*). Of the double mutants, the *bm3-bm4* mutant showed dramatically

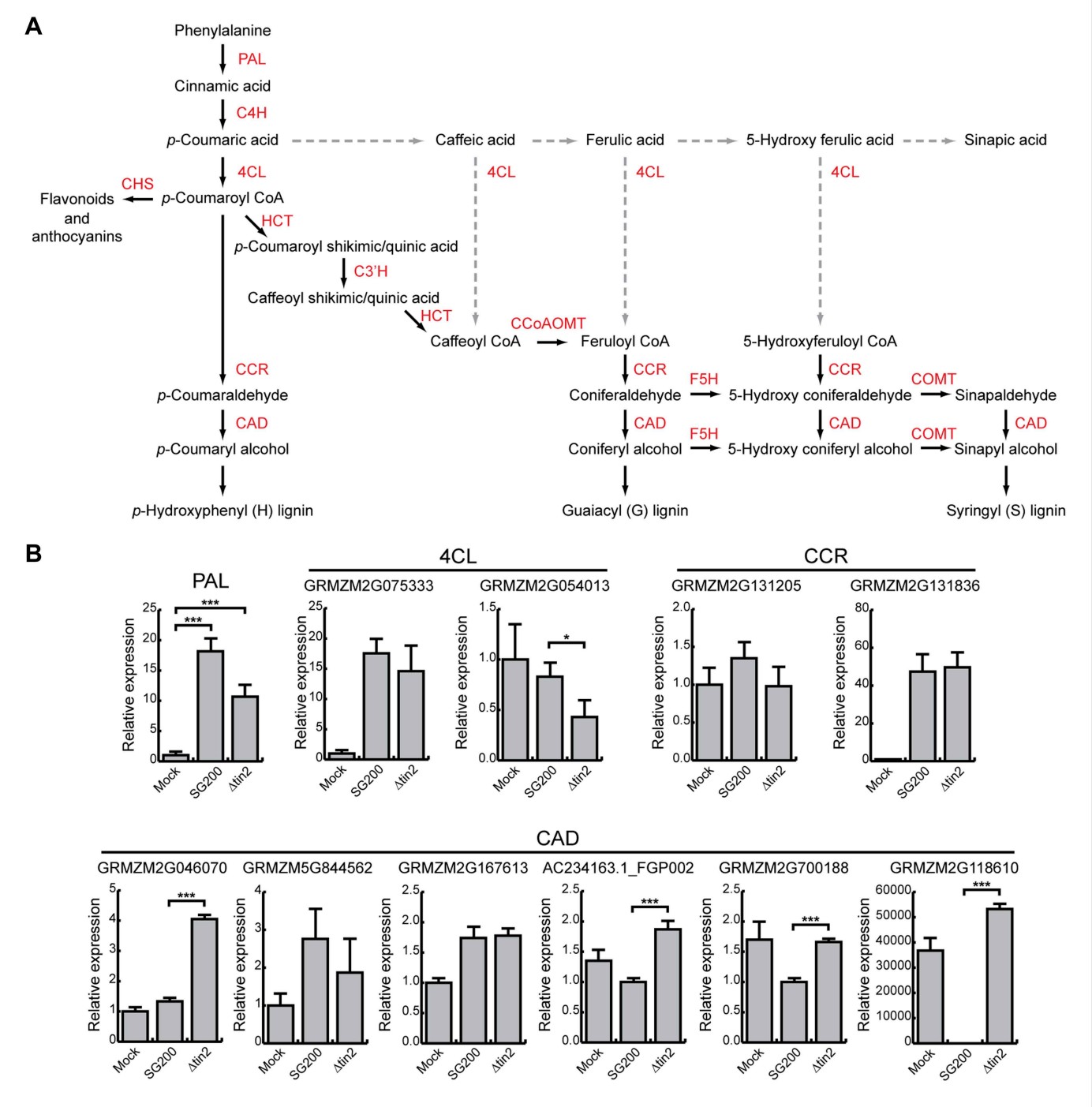

**Figure 12**. Phenylpropanoid pathway genes are differentially expressed after infection with SG200 and SG200Δtin2. (**A**) Schematic model of the phenylpropanoid pathway in maize. The enzymes involved in each biosynthetic step are shown in red. PAL, phenylalanine ammonia lyase; C4H, cinnamate 4-hydroxylase; 4CL, 4-coumarate-CoA ligase; HCT, hydroxycinnamoyl transferase; C3'H, p-coumaroyl shikimate 3'-hydroxylase; CCoAOMT, caffeoyl CoA 3-*O*-methyltransferase; F5H, ferulate 5-hydroxylase; COMT, caffeic acid 3-*O*-methyltransferase; CCR, cinnamoyl-CoA reductase; CAD, cinnamyl alcohol dehydrogenase; CHS, chalcone synthase. Dashed lines indicate routes, which may occur in maize under certain conditions, but are not considered major routes based on experimental evidence (***Humphreys and Chapple, 2002***). (**B**) qPCR based quantification of the expression levels of genes from the phenylpropanoid pathway after infection with indicated *U. maydis* strains and collecting infected leaf material 6 dpi. *p < 0.05, ***p < 0.001.

The following source data are available for figure 12:

**Source data 1**. List of upregulated maize genes after SG200Δtin2 infection (vs SG200).

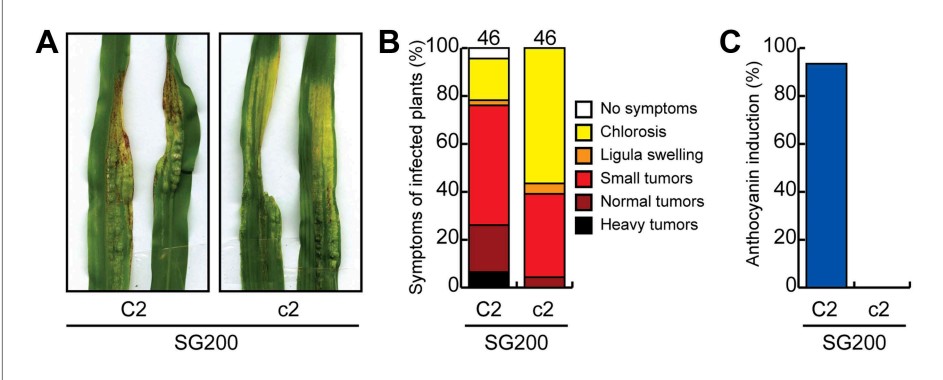

**Figure 13**. Pathogenicity of *U. maydis* in maize lacking chalcone synthase. (**A**) Macroscopic SG200 disease symptoms on C2 (control) and c2 (chalcone synthase mutant) after 12 dpi are shown. (**B**) Disease symptoms were scored in two independent SG200 infections using the scoring scheme depicted on the right and the data were combined. (**C**) From the plants scored in (**B**) the percentage of plants showing anthocyanin induction is given.

enhanced disease symptoms, while disease symptoms in the *bm2-bm3* double mutant was lower than in the single *bm3* mutant (***Figure 14C***), suggesting a negative effect of the *bm2* mutation on virulence in this combination. When *bm2* was combined with *bm1*, virulence was comparable to the single *bm1* mutant (***Figure 14C***). The alterations in virulence in the *bm* mutants were not only visible in symptom severity but also with respect to symptom distribution (***Figure 14D***). After infection with SG200, more than 80% of leaf tumors stayed locally confined, while in the *bm* mutants (with the exception of the single *bm2* mutant) leaf tumors were much more spread and covered the entire leaf area from the injection hole to the stem (***Figure 14B,D***), suggesting that lignification restricts *U. maydis* growth and spread in the infected tissue. In line with this assertion, we observed strongly lignified cells in vascular bundle tissue in cross sections of maize leaves infected with SG200Δtin2 and could only rarely observe fungal hyphae in bundle sheath cells (***Figure 14E***). By contrast, lignification was significantly less pronounced in SG200-infected tissue (***Figure 14F***) and in tissue infected by SG200Δtin3 (***Figure 14G***), a mutant lacking another effector from cluster 19A that does not affect anthocyanin induction (***Brefort et al., in press***). Additionally, SG200 and SG200Δtin3 hyphae were readily detected inside bundle sheath cells (***Figure 14F,G***). This suggests that anthocyanin induction by the Tin2 effector attenuates lignification of vascular bundle cells, conceivably facilitating fungal access to nutrients in vascular bundle tissue.

## Discussion

Comprehensive genetic, biochemical and transcriptome analyses allowed us to uncover a connection between anthocyanin and lignin biosynthesis that is targeted by a novel secreted effector of *U. maydis*, Tin2. Our results are summarized in the model depicted in ***Figure 15***. The link between Tin2 and anthocyanin biosynthesis appears direct and occurs through the stabilization of the kinase ZmTTK1 via Tin2. We have shown that after *U. maydis* infection, when Tin2 is present, the stabilization of ZmTTK1 will increase biosynthesis of anthocyanin. Conversely, when *U. maydis* is lacking *tin2*, more of the common precursor for anthocyanin and lignin, 4-coumaric acid, is likely to be available for lignin biosynthesis (***Figure 15***) resulting in cell wall fortification in vascular tissue. The deposition of lignin in the plant cell wall is considered to provide an undegradable physical barrier to infection (***Vance et al., 1980***). The observed decrease in ferulic acid in SG200Δtin2-infected leaves compared to SG200-infected tissue is consistent with increased lignification where ferulic acid acts as a nucleation site for lignin polymerization (***Grabber et al., 2002***). Tin2 affected all steps of the anthocyanin pathway, but did not affect transcription of all genes coding for lignin pathway components. While none of the early genes implicated in lignin biosynthesis were upregulated after infection with *U. maydis* wild type, we observed transcriptional effects for four of the six *CAD* genes acting at a late step of the lignin pathway (***Figure 12A***). At present, there is only a limited understanding of how the lignin biosynthetic pathway in maize is regulated. Several Myb and NAC transcriptional activators involved in secondary cell wall formation have been identified (***Fornalé et al., 2010***; ***Zhong et al., 2011***), but the mechanisms behind

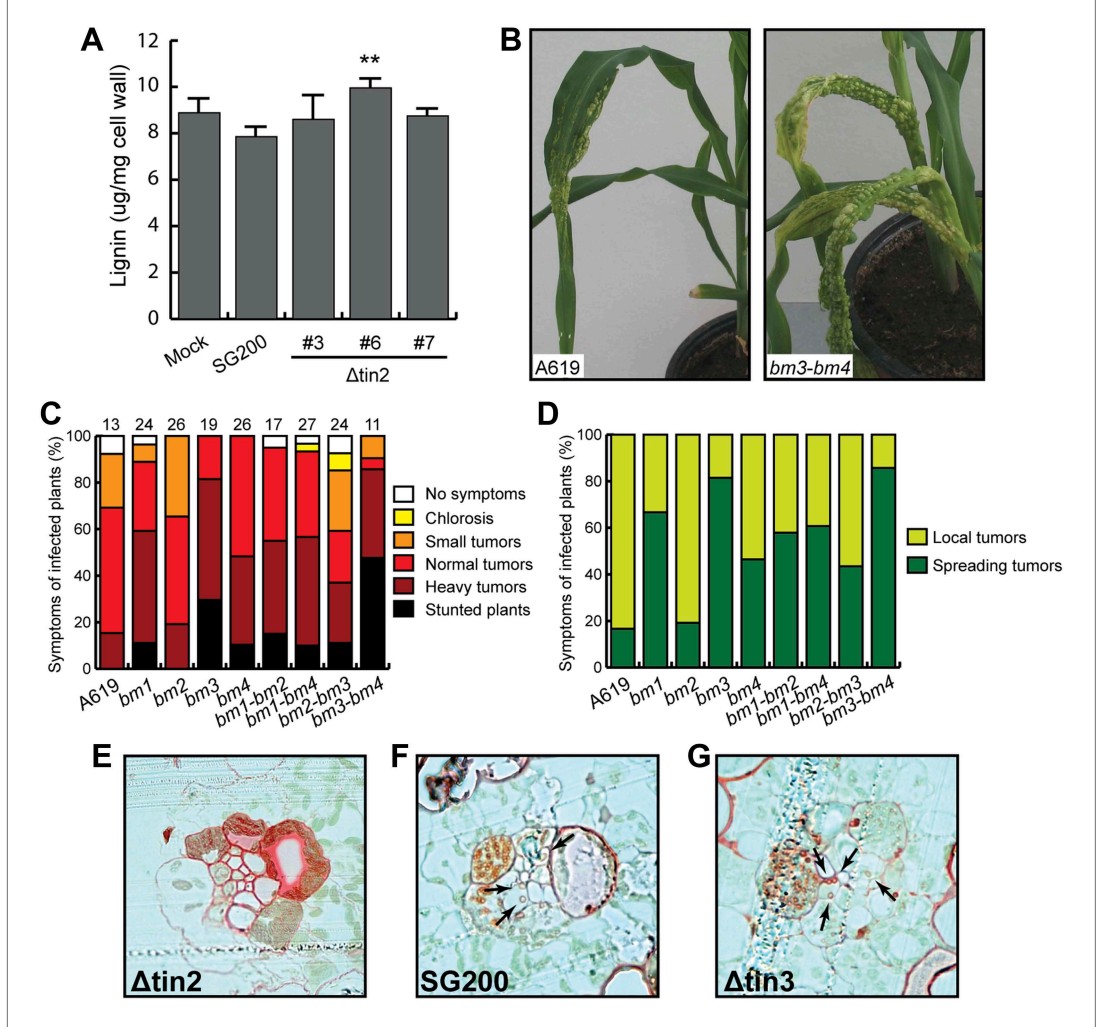

**Figure 14**. Virulence of *U. maydis* in hosts with altered lignin. (**A**) Measurement of total lignin content in leaves infected with *U. maydis.* Infected leaf segments from mock inoculated leaves and leaves infected with three independent SG200Δtin2 strains as well as SG200 were collected at 6 dpi. Excised leaf segments were ground in liquid nitrogen and extracted with methanol. Isolated lignin was hydrolyzed under alkaline conditions and amount of monomers was measured at 280 nm using coniferyl alcohol for calibration. Error bars indicate the standard deviation of three biological replicates. **p<0.01 (vs SG200). (**B**) Macroscopic disease symptoms of *U. maydis* SG200 on maize A619 and the double *brown midrib* mutant *bm3-bm4.* Symptoms were scored at 12 dpi. (**C**) Pathogenicity of *U. maydis* SG200 on indicated *bm* mutants. Virulence was scored at 12 dpi. The total number of infected plants is indicated above each panel and data were combined from two independent experiments. (**D**) Symptoms of SG200 on *bm* mutants shown in (**C**) were classified into local tumors (tumors are restricted in one area on the leaf blade), and spreading tumors (tumors extend from infected leaf area all the way into the stem area). (**E**–**G**) Lignin deposition in cross-sections of plants infected with the indicated strains was visualized by Safranin O staining (red). Fungal hyphae inside vascular bundles are indicated by arrows.

tissue-specific and developmentally regulated gene expression, and the role of posttranscriptional regulation of gene expression are yet to be fully elucidated. In *tin2* mutant infected tissue total lignin levels appear increased, although statistically relevant effects were seen in only one of the mutants analyzed (*Figure 14A*). This may reflect that increased lignification does not affect the entire leaf but, as we have shown, is restricted to the vascular tissue colonized by the *tin2* mutant late in infection. Therefore, when total tissue is analyzed, differences in total lignin content may not be so evident in wild-type and *tin2* mutant infected tissue. However, the increased susceptibility of maize *bm* mutants to *U. maydis* and the increased leaf areas displaying disease symptoms, provide strong evidence for a role of lignification in disease resistance. Several mechanisms may form the basis for this observation. A less lignified cell wall will be easier to penetrate by the fungus. The plant may also be less effective at producing defense-related lignin that could slow down the migration of the fungus. Lastly, since the

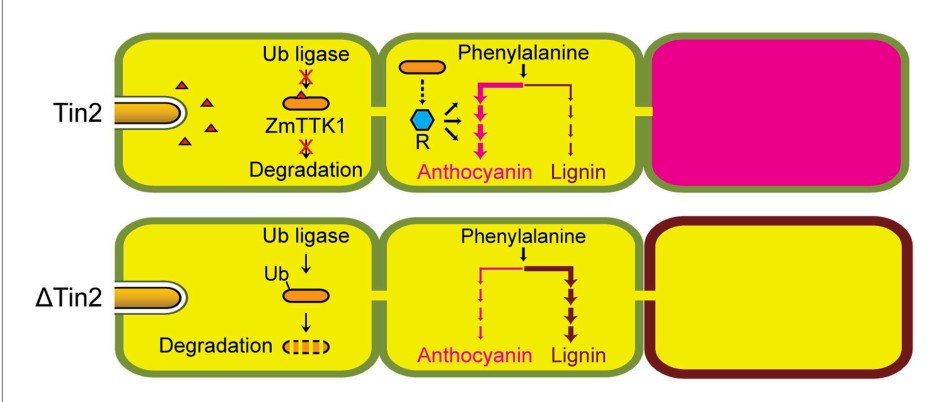

**Figure 15**. Hypothetical model for Tin2 function. In the upper part the hypothetical model for infection of maize with wild-type *U. maydis* is depicted. Tin2 effector (red triangles) is secreted, taken up by maize cells and binding to the ZmTTK1 maize kinase (orange ellipsoid), which leads to its stabilization and renders it active. Active kinase is proposed to positively affect anthocyanin biosynthesis (dark pink) via the R transcription factor. Arrows and their thickness depict the activity of the anthocyanin and lignin pathway, respectively. The lower part depicts the hypothetical model for infection with an *U. maydis tin2* mutant. In the absence of Tin2, ZmTTK1 is degraded via the ubiquitin-proteasome system. As a result, there is no anthocyanin biosynthesis, which could conceivably allow more precursor to enter the lignin branch, leading to fortified cell walls (brown lining) that limit fungal access.

fungus spreads via the vascular tissue and may rely on nutrients obtained from this tissue for massive proliferation, the altered physico-chemical characteristics of the vascular tissue in the *bm* mutants, combined with altered dimensions of the xylem vessels (***Vermerris et al., 2010***) may influence entry and the rate of migration of the fungus. The *bm2* mutant appears to be an exception in that it does not show increased susceptibility to *U. maydis*, and even appears to attenuate the effect of infection in *bm1-bm2* and *bm2-bm3* double mutants relative to the *bm1* and *bm3* single mutants. This effect may be due to the accumulation of a fungitoxic (phenolic) compound in this mutant, as has been proposed for the sorghum *brown midrib12* mutant, which was shown to be less susceptible to colonization by *Fusarium* spp. (***Funnell-Harris et al., 2010***). Also in *N. attenuata* lines in which two *CAD* genes were silenced, compensatory processes for the structural deficiencies were observed when plants were grown in the field, and in this case such plants accumulated various phenylpropanoids (***Kaur et al., 2012***). Therefore, while we consider direct effects of Tin2 on lignification to be the simplest model to explain the reduced virulence on the *tin2* mutant strain (***Figure 15***), the situation may in reality be much more complex. For example we cannot formally rule out that in the absence of anthocyanin biosynthesis it is not actually the enhanced lignification that affects fungal spread but an altered composition of the plant cell walls. Conversely, the enhanced disease symptoms observed in the *bm* mutants could also be attributed to an lower incorporation of phenolics rather than to the reduction in lignin, that is in wild-type plants cell wall phenolics may negatively affect fungal growth and restrict pathogen spread.

Reduction in lignin content and changes in lignin subunit composition have been shown to improve biomass conversion in a number of species (***Chen and Dixon, 2007***; ***Studer et al., 2011***), including maize, with the *bm1* and *bm3* mutations leading to 50% higher yields of fermentable sugars on a per-gram-biomass-basis (***Vermerris et al., 2007***). Based on the experiments described here, it is clear that modifying lignin composition in maize has the risk of increased susceptibility to *U. maydis*, and the associated reduction in yield.

Whether Tin2 is actively diverting the flux from lignin to anthocyanin remains speculative as flux measurements would require the absolute quantification of lignin and anthocyanin in the infected tissue and this is hampered by the technical difficulties in lignin extraction and quantification due to limited amounts of infected tissue material. In addition, the direct or indirect effects of Tin2 on other pathways like DIMBOA and flavonoid biosynthesis might complicate such analyses. However, precedence for a connection between lignin and anthocyanin biosynthesis comes from *A. thaliana*, where the silencing of HCT (hydroxycinnamoyl transferase), the first committed enzyme from the lignin pathway,

redirected the metabolic flux into flavonoids through chalcone synthase and this resulted in elevated levels of flavonols and anthocyanin (*Besseau et al., 2007*). In addition, transgenic *A. thaliana* plants expressing the lignin biosynthesis repressing ZmMYB31 transcription factor from maize redirected carbon flux towards the biosynthesis of anthocyanins (*Fornalé et al., 2010*). Recently, a competition between anthocyanin and lignin pathways for their common precursor has also been detected in strawberry (*Ring et al., 2013*). In that system it was demonstrated that plant class III peroxidase (FaPRX27), a gene connected with lignin biosynthesis, is linked to a region implied in fruit color decrease. In addition, it was demonstrated that the downregulation of chalcone synthase lead to an induction of FaPRX27 and this diverted the flux from anthocyanins to lignin (*Ring et al., 2013*). Thus, there appears to be a metabolic connection between anthocyanin and lignin biosynthesis pathways in monocot as well as dicot systems and *U. maydis* may be altering this with the help of Tin2. Future work will be needed to address whether this connection is direct with lignin functioning as barrier or indirect with anthocyanin induction negatively affecting the accumulation of other defense compounds.

Transient expression of the ZmR1 transcription factor known to positively regulate the anthocyanin pathway in maize revealed strong nuclear localization when ZmR1 was co-expressed with ZmTTK1 and Tin2, but not when Tin2 was omitted. We suggest that the difference in ZmR1 localization may be caused by its phosphorylation, although we cannot exclude other possibilities. In *A. thaliana* MYB75, which is related to maize ZmC1, but is not the closest ortholog, acts as a repressor for the lignin-specific branch of the phenylpropanoid pathway presumably through an interaction with TT8, a bHLH protein related to ZmR1 (*Bhargava et al., 2010*). If such a scenario also exists in maize, the observed Tin2-mediated nuclear localization of ZmR1 might actually promote the formation of such a repressive complex for lignin biosynthesis. How Tin2 affects the expression of DIMBOA is currently unclear. It is conceivable that this could reflect a co-regulation with anthocyanin biosynthesis genes. Alternatively, the activity of another transcription factor might be modulated by the Tin2-stabilized ZmTTK1 protein. An upregulation of *Bx1* and elevated amounts of DIMBOA had previously been detected after colonization of maize by *U. maydis* (*Basse, 2005*). In this context, it had also been demonstrated that *U. maydis* is resistant to DIMBOA (*Basse, 2005*) while several other fungal pathogens of maize are susceptible (*Glenn et al., 2002*). This could indicate that the induction of DIMBOA might deter infection by other pathogens in *U. maydis* infected tissue. However, in a macro-array based expression analysis of phenylpropanoid and related genes in *brown midrib* mutants, two genes from the DIMBOA pathway, *Bx4* and *Bx5*, were significantly upregulated in the *bm3* mutant, that is the mutant with the strongest effect in lignin biosynthesis (*Guillaumie et al., 2007*). The authors suggest that this may be a compensatory reaction for the lack of COMT activity in the *bm3* mutant (*Guillaumie et al., 2007*). If true, the upregulation of DIMBOA after SG200 infection could also be the consequence of the locally reduced lignin content in the infected areas.

For its function, the Tin2 protein must translocate into plant cells after being secreted. Although an mCherry-Tin2 fusion protein produced by *U. maydis* was partially biologically active, we could never see the fusion protein accumulate inside maize cells. Based on similar results with Cmu1, another transferred effector (*Djamei et al., 2011*), we speculate that large fluorescent tags fused to the respective effector prevent uptake in this pathosystem. Cmu1 and Tin2 do not share obvious common motifs that can be recognized in their primary amino acid sequence that could serve as translocation signal, and it remains to be determined how they cross the plant plasma membrane. Anthocyanin induction after transient transformation of Tin2 could be observed only when plants were pre-infected with a *tin2* mutant strain. We consider that it may be necessary to pre-infect plants to obtain high levels of phenylalanine or tyrosine for anthocyanin biosynthesis. These amino acids supply the flavonoid pathway and *U. maydis* infection was shown to increase their amounts in infected leaves through Cmu1 and possibly other effectors (*Doehlemann et al., 2008*; *Djamei et al., 2011*).

In maize, anthocyanin induction is connected to kernel development (*Holton and Cornish, 1995*) and is also observed when coleoptiles are exposed to UV-B (*Beggs and Wellman, 1985*). We hypothesize that there will be a native regulatory system stabilizing the ZmTTK1 protein kinase in such conditions, and Tin2 may mimic this regulator. Interestingly, ZmTTK1 orthologs in barley and *B. distachyon* have substitutions in critical serine residues of the phosphodegron motif DSGxS, and in the related kinase from *A. thaliana* this region is missing altogether (*Figure 7A*). Since the barley smut *U. hordei* and the *B. distachyon* smut *U. bromivora* (*Barbieri et al., 2011*) both lack an ortholog of Tin2 (*Laurie et al., 2012*; AD and RK, unpublished data), this could suggest that the ongoing co-evolution between

host and pathogen has already generated stable kinases in barley and *B. distachyon*, respectively, that no longer need respective orthologs of the Tin2 effector for activity.

Anthocyanin serves for pollinator attraction of flowers (*Sheehan et al., 2012*). In addition, its accumulation is frequently observed under abiotic and biotic stress conditions. Anthocyanin is transported and accumulates in the vacuole (*Goodman et al., 2004*), provides UV-B stress protection and allows scavenging of reactive oxygen species (*Steyn et al., 2002*). So far, the observed accumulation of anthocyanin after biotic stress was discussed in the frame of defense responses of resistant cultivars (*Hammerschmidt and Nicholson, 1977*; *Hipskind et al., 1996*). In addition, some other examples, not connected to classical resistance responses, have been described: in a recent report it has been demonstrated that tomatoes enriched in anthocyanins are less susceptible to the necrotrophic fungus *Botrytis cinerea*. The enhanced resistance occurs through reduced spreading of the ROS burst, which limits the induction of cell death necessary for growth of this necrotroph (*Zhang et al., 2013*). In potato it was shown that ectopic overexpression of an anthocyanin 5-*O*-glycosyl transferase conferred enhanced resistance to the bacterial pathogen *Erwinia carotovora*. In this system, it is not clear, however, whether the effects are direct or indirect (*Lorenc-Kukuła et al., 2005*). In perennial ryegrass infected with the fungal endophyte *Epichloae festucae*, prominent anthocyanin induction is observed at the base of tillers, while fungal Δ*sak* mutants, lacking a stress activated MAP kinase, fail to induce anthocyanin, display alterations in the vascular tissue and accumulate less fungal biomass (*Eaton et al., 2010*). In this system it has been speculated that the observed downregulation of key enzymes in the shikimate and anthocyanin pathway may divert precursors into the catechin/tannin pathway with negative consequences for fungal development (*Eaton et al., 2010*). In the present study, we have uncovered a new positive connection between anthocyanin induction and development of a biotrophic pathogen. We suggest that examples where anthocyanin induction is observed in response to biotic stress should be revisited to see, whether metabolic rewiring of the phenylpropanoid pathway is adopted as a common strategy by biotrophic microbes colonizing plants.

## Materials and methods

### Growth conditions and virulence assays

*Zea mays* cv. Early Golden Bantam (Olds Seeds, Madison, WI, USA) was grown in a temperature-controlled greenhouse (14 hr/10 hr light/dark cycle; 28°C/20°C) and used for infection by *U. maydis*. *Z. mays* chalcone synthase mutant M541J (A1 A2 C1 *c2* R1-nj) and cognate control strain M142V (A1 A2 C1 C2 R1-nj) were obtained from the Maize Genetics Stock Center (http://maizecoop.cropsci.uiuc.edu/) and are near-isogenic lines. The *brown midrib* maize mutants *bm1*, *bm2*, *bm3* and *bm4* mutant are near-isogenic lines generated in the background of inbred line A619 (*Vermerris and McIntyre, 1999*; *Marita et al., 2003*), following eight backcrosses. The double mutants *bm1-bm2*, *bm1-bm4*, *bm2-bm3* and *bm3-bm4* were created by crossing the respective single mutant lines, followed by self-pollination of the resulting $F_1$ plants, selection of double mutants among the $F_2$ progeny, and confirmed with test crosses to the individual single mutants (*Vermerris et al., 2010*). *U. maydis* strains were grown in YEPSL (0.4% yeast extract, 0.4% peptone, 2% sucrose) and cell suspensions in $H_2O$ adjusted to $OD_{600} = 1.0$ were injected into the stem of 7-day-old maize seedlings with a syringe as described (*Kämper et al., 2006*). Disease symptoms were scored at 12 days post infection using a previously developed scoring scheme (*Kämper et al., 2006*) and statistical analysis was performed as described (*Brefort et al., in press*). *N. benthamiana* was grown under the same conditions as maize.

### Plasmid constructs and mutant generation

For plasmid constructions, standard molecular cloning strategies and techniques were applied (*Sambrook et al., 1989*). All plasmids used and generated are listed in *Supplementary file 1*. To generate truncated *tin2* genes, the plasmid pIP-pum05302 (*Brefort et al., in press*) was digested with *Xba*I-*Asc*I. Truncated *tin2* genes were amplified by respective primer pairs (*Supplementary file 2*) and introduced into the *Xba*I-*Asc*I site. $tin2_{AAAAA}$ was generated by amplification with respective primers (*Supplementary file 2*) and used to replace *um05302* in pIP-pum05302. For constitutive overexpression of *tin2* in *U. maydis*, p123P$_{otef}$ containing the constitutive *otef* promoter (*Wang et al., 2011*) was digested with *Nco*I-*Not*I and full-length *tin2* amplified by respective primer pairs (*Supplementary file 2*) was inserted. For transient expression in maize cells, p35S-mCherry (*Djamei et al., 2011*) was digested by *Eco*RI-*Xba*I and *tin2* or ZmTTK1 genes amplified by respective primers (*Supplementary file 2*) were introduced.

For yeast two-hybrid constructs, pGBKT7 or pGADT7 were digested with *Nde*I-*Bam*HI or *Eco*RI-*Xho*I, respectively and *tin2* and truncated *tin2* genes or ZmTTK1 and truncated versions thereof were inserted after amplification with respective primer pairs (*Supplementary file 2*). For protein expression in *N. benthamiana*, *tin2*, and ZmTTK1 were amplified by respective primer pairs (*Supplementary file 2*) and inserted into pBIN19AN (*Hirota et al., 2007*) digested with *Apa*I-*Not*I. Point-mutations in ZmTTK1 were introduced using the QuikChange Site-Directed Mutagenesis Kit (Agilent Technologies, Santa Clara, USA) following the manufacturer's protocol using primers (*Supplementary file 2*). All plasmid constructs containing amplified gene fragments were sequenced.

The haploid solopathogenic strain SG200 of *U. maydis* (*Kämper et al., 2006*) was used as a reference strain throughout this study. SG200Δ19A-1, SG200Δtin2, and SG200Δtin3 mutants were generated in a previous study (*Brefort et al., in press*). All *U. maydis* strains (*Supplementary file 3*) are derived from SG200 and were generated by insertion of p123 derivatives into the *ip* locus as described (*Loubradou et al., 2001*). Isolated *U. maydis* transformants were tested by southern analysis to assess single or multiple integration events in the *ip* locus.

## Analysis of anthocyanins, flavonoids and DIBOA-glucoside

Maize seedlings inoculated with $H_2O$, SG200, SG200Δtin2 and SG200Δtin2-tin2 were harvested 6 days after infection. 2–3 cm segments below the injection hole showing anthocyanin induction in leaves infected with SG200 and SG200Δtin2-tin2 were excised from 20 plants, pooled and ground in liquid nitrogen. From plants infected with the *tin2* mutant strains corresponding leaf segments, which did not show anthocyanin pigmentation, were excised.

Total anthocyanins were quantified according to the method described by *Ray et al. (2003)*. 100 mg homogenized plant material was heated for 10 min in 1 ml of 2 N HCl at 55°C, cooled down and incubated overnight in the dark at room temperature. The extract was centrifuged and the clear supernatants were analyzed at 515 nm at a Cary 100 Bio Spectrophotometer (Varian, Darmstadt, Germany). Commercial cyanidin 3-arabinoside chloride (PhytoLab, Vestenbergsgreuth, Germany) was used for quantification. For further identification of anthocyanins, flavonoids, and DIBOA-glucoside, as well as for recording their profiles, powdered samples were extracted and the polar phase was measured by UPLC-PDA-TOF-MS as described (*Djamei et al., 2011*). For unambiguous identification fragmentation studies were performed by Ultra High Performance Liquid Chromatography (1290 Infinity, Agilent Technologies) coupled with a 6540 UHD Accurate-Mass QTOF LC MS instrument with Agilent Jet Stream Technology as ESI source (Agilent Technologies) as described (*Koch et al., 2013*).

## Lignin extraction and analysis

For cell wall isolation, 200 mg of plant material from the same batch was used for the anthocyanin quantification, was extracted with 1.6 ml 80% methanol (vol/vol) for 1 hr in a 2-ml plastic tube and centrifuged for 20 min at 16,000×*g*. The pellet was washed two times with 1.6 ml of 1 M NaCl and 0.5% Triton X-100, for two times with 1.8 ml water, two times with 1.7 ml of methanol, and two times with 1.7 ml of acetone. Next alkaline hydrolysis was performed by adding 1.7 ml 2 M NaOH followed by incubation for 1 hr under vigorous shaking. Then, the pellet was washed twice with 1.8 ml water and dried at 80°C overnight. For lignin determination, 2 mg of the cell wall material was used and 250 µl of 25% (vol/vol) acetylbromide in glacial acetic acid were added and incubated for 30 min at 70°C. Afterwards, the sample was cooled down immediately on ice and 250 µl 2 M NaOH were added before centrifuging for 5 min at 16,000×*g*. For UV analysis (280 nm), 139 ml of the supernatant were taken and 2.8 µl 50% (vol/vol) hydroxylamine and 1.25 ml glacial acetic acid were added. A calibration curve was generated with coniferyl alcohol as standard compound in the range from 5 to 20 µg.

## Bioinformatic tools applied in this study

Signal peptide prediction was performed with the online program SignalP 4.1 (http://www.cbs.dtu.dk/services/SignalP/). Protein secondary structure prediction was performed with the online program Jpred3 (http://www.compbio.dundee.ac.uk/www-jpred/). Phylogenic analysis was performed with the online program Phylogeny.fr (http://www.phylogeny.fr/). Protein conserved domain search was performed with the online program Pfam (http://pfam.sanger.ac.uk/search/).

## Microarray analysis

For the microarray experiments, maize plants (Early Golden Bantam) were grown in a phytochamber in a 15 hr/9 hr light/dark cycle; light period started/ended with 1 hr ramping of light intensity. Temperature

was 28°C and 20°C, relative humidity 40% and 60% during light and dark periods, respectively, with 1 hr ramping for both values. Plants were inoculated with $H_2O$ (mock), SG200, and SG200Δtin2 as described above. Infected or water-inoculated tissue from 20 plants per experiment was harvested at 4 dpi by excising a section of the third leaf between 1 and 3 cm below the injection holes. For RNA extraction, material was pooled, ground to powder under liquid nitrogen and RNA was extracted with Trizol reagent (Invitrogen, Karlsruhe, Germany). RNA was purified applying the RNeasy kit (Qiagen, Hilden, Germany). Affymetrix maize genome arrays were done in three biological replicates, using standard Affymetrix protocols (Midi_Euk2V3 protocol on GeneChip Fluidics Station 450; scanning on Affymetrix GSC3000G). Expression data were submitted to Gene Expression Omnibus (http://www.ncbi.nlm.nih.gov/geo/) (Accession number: GSE48536). The microarray data obtained in this study were analyzed using the Partek Genomics Suite version 6.12. Expression values were normalized using the RMA method. Criteria for significance were a corrected p-value (per sample) with an FDR of 0.05 and a fold-change of >2. Differentially expressed genes were calculated by a 1-way ANOVA.

## Quantitative real-time PCR and fungal biomass analyses

Total RNA was extracted from *U. maydis* cell grown in YEPSL medium and from infected leaves by excising 2–3 cm segments from below the injection holes. At least 20 leaf segments were pooled for each of the indicated time points. RNA was isolated with the Trizol reagent (Invitrogen) following manufacturer's protocol. cDNA was synthesized with Superscript III First Strand Synthesis SuperMix (Invitrogen). Quantitative real-time PCR reactions were performed as previously described (*Berndt et al., 2010*). All reactions were performed in at least three biological replicates. Statistical analysis was performed with Student's *t*-test. Relative expression levels of fungal or maize genes were calculated in relation to the values obtained for the constitutively expressed peptidylprolyl isomerase gene (*ppi*) of *U. maydis* or glyceraldehyde 3-phosphate dehydrogenase (*GAPDH*) of *Z. mays*. Primers used for PCR are listed in *Supplementary file 2*. For fungal biomass assessment, samples from infected plants were prepared as above. DNA was extracted and fungal biomass was determined as previously described (*Brefort et al., in press*).

## Yeast two hybrid interaction assays

To screen for interactors of Tin2, Tin2 lacking the signal peptide for secretion was expressed from pGBKT7-Tin2$_{26-207}$ and screened against a pAD_Gal4-library constructed from cDNAs generated from RNA of *U. maydis* FB1 × FB2 infected tissue. Material was collected 2 and 5 days after infection (*Farfsing, 2004*). Transformation of *Saccharomyces cerevisiae* strain AH109 (Clontech, Mountain View, USA) was done as described in the DUAL membrane starter kit manual (Dualsystems Biotech AG, Schlieren, Switzerland). The yeast two-hybrid screen was performed following the instructions of the matchmaker yeast two-hybrid manual (Clontech) using 1 mg of bait-DNA (pGBKT7-Tin2$_{26-207}$) and 0.5 mg of library-DNA. All resulting yeast clones were tested by immunoblotting for expression of the respective fusion proteins. Interacting clones were verified by empty vector controls excluding self-activating clones as well as by vector swap experiments. Full-length cDNAs of putative interaction partners were generated from SG200-infected tissue collected 6 days post infection, cloned in pGADT7 and tested for interaction with Tin2$_{26-207}$ expressed from pGBKT7-Tin2$_{26-207}$.

For yeast protein interaction assays, the genes encoding the proteins tested for interaction were cloned into pGBKT7 or pGADT7 vectors (Clontech), respectively, to express fusion proteins with the yeast GAL4 binding (BD) and activation domain (AD), respectively, in strain AH109 (Clontech). Yeast transformants were screened on selective dropout media (SD) lacking only tryptophan (Trp) and leucine (Leu) to select yeast cells, which contained the desired plasmids (*Supplementary file 4*). Protein interactions were assessed on SD selection medium lacking tryptophan, leucine, and histidine (His) containing 3 mM 3-Amino-1, 2, 4-triazole (3-AT). Yeast MEL1p activity was measured by addition of 5-bromo-4-chloro-3-indolyl α-D-galactopyranoside (X-α-gal). Fusion protein expression was verified by western blot.

## Protein expression and protein in vitro studies

For protein expression in *Escherichia coli*, pRSET-GST-PP (*Mueller et al., 2013*) and pPR-IBA102 (IBA Lifesciences, Göttingen, Germany) vectors were used for the preparation of GST-Tin2$_{26-207}$ or Strep-ZmTTK1, respectively. *E. coli* BL21 (DE3) strain was used as host for protein expression. *E. coli* transformants carrying respective expression plasmids were grown in YT medium (1.6% Bacto Tryptone, 1.0% Yeast Extract, 0.5% NaCl) containing ampicillin (50 µg/ml). Protein induction was performed by

addition of 1 mM isopropyl β-D-1-thiogalactopyranoside (IPTG) at 28°C for 5 hr. Cell pellets were lysed by BugBuster master mix (Merck, Schwalbach, Germany). Soluble supernatant was incubated with Glutathione Sepharose 4 Fast Flow (GE Healthcare, München, Germany) or Strep-tactin sepharose (IBA Lifesciences) for GST- or Strep-tagged protein purification, respectively. Protein enrichment and purity was assessed by western blot. For in vitro co-IP, recombinant GST-Tin2$_{26-207}$ or GST-Tin2$_{26-202}$ protein (200 µg) bound to glutathione sepharose beads (GE healthcare), respectively, was incubated with 500 µl $E.\ coli$ soluble extract in PBS buffer (10% glycerol, 0.1% Tween20 and 0.5 mg/ml bovine serum albumin) from induced BL21(DE3)/pPRIBA102-ZmTTK1 at 4°C for 1 hr. GST fusion proteins were eluted with reduced glutathione. Strep-ZmTTK1 in eluate was detected by western blot. Detection of Tin2-HA protein in culture supernatants of $U.\ maydis$ was performed as previously described (**Doehlemann et al., 2009**).

For western blot analysis, protein was separated by 10% or 13% SDS-polyacrylamide gel electrophoresis. Protein was visualized by staining with Coomassie Brilliant Blue (CBB). Monoclonal anti-c-myc antibody produced in mouse, anti-HA antibody produced in rabbit or monoclonal anti-ubiquitin antibody produced in mouse were purchased from Sigma-Aldrich (Taufkirchen, Germany) and used as a primary antibody. Anti-mouse IgG HRP-linked or anti-rabbit IgG HRP-linked antibodies were purchased from Cell Signaling (Danvers, MA, USA) and used as a secondary antibody. Strep-tactin-HRP (IBA Lifesciences) was used for the detection of Strep-tagged protein. For immunoprecipitation of ZmTTK1 from plant cell, total plant protein extract was incubated with human-IgG agarose (Sigma-Aldrich) at 4°C for 2 hr on a rotary shaker. For in vitro kinase assay, recombinant proteins or human-IgG agarose carrying immunoprecipitates were incubated at 28°C for 30 min in a volume of 25 µl reaction buffer containing 0.5 mg/ml myelin basic protein (Sigma-Aldrich), 20 µM ATP, and 2.5 µCi [γ-$^{32}$P]ATP. Phosphorylated myelin basic protein was detected using a STORM phosphoimager (GE Healthcare).

## Transient protein expression in *Z. mays* and *N. benthamiana*

For protein expression in *N. benthamiana*, *Agrobacterium tumefaciens* GV3101 strains carrying expression plasmids indicated in each experiment were grown in YT medium containing kanamycin (50 µg/ml). Cells were resuspended in 10 mM MES (pH 5.6) and the concentration was adjusted as OD$_{600}$ = 0.5. *A. tumefaciens* strains were mixed at an equal ratio with the *A. tumefaciens* strain LBA2566 expressing the 35S:p19 construct encoding the p19 silencing suppressor (**Voinnet et al., 2003**). Bacterial mixtures were infiltrated with needleless syringe to the upper leaves of 3-weeks-old *N. benthamiana* plants. Protein was extracted from the leaves at 3 days after infiltration with extraction buffer (100 mM Tris-HCl pH 7.5, 100 mM NaCl, 5 mM EDTA, 5 mM EGTA, 10 mM NaF, 0.1 mM Na$_3$VO$_4$, 10 mM β-glycerophosphate, 1 mM β-mercaptoethanol, 0.1% Triton-X100, 10% Glycerol, protease inhibitor cocktail [Roche, Mannheim, Germany]) under liquid nitrogen. For proteasome inhibition, 100 µM MG132 (Merck) solution was infiltrated into *N. benthamiana* leaves at 1 day before protein extraction.

For transient protein expression in maize by biolistic gene transfer, 1.6 µm gold particles (Bio-Rad, München, Germany) were coated with the plasmids coding for the indicated genes driven under CaMV 35S promoter (**Supplementary file 1**). Bombardment was performed using a PDS-1000/He system (Bio-Rad) onto 2-weeks-old maize leaves or SG200Δtin2-infected leaves (at 4–5 dpi).

## Microscopic observation

To visualize fungal proliferation in infected tissue, the area 1–3 cm below the injection site was excised at 3 days post infection. Fungal hyphae were stained with WGA-AF488 (Invitrogen), the plant cell wall was stained with propidium iodide (Sigma-Aldrich). Leaf samples were incubated in staining solution (1 µg/ml propidium iodide, 10 µg/ml WGA-AF488) and observed with a TCS-SP5 confocal laser-scanning microscope (Leica Microsystems, Wetzlar, Germany) under the following conditions. WGA-AF488: excitation at 488 nm and detection at 500–540 nm, propidium iodide: excitation at 561 nm and detection at 580–660 nm. For fluorescent protein detection, leaf samples were directly observed by confocal microscopy under the following conditions. mCherry: excitation at 561 nm and detection at 580–630 nm, YFP: excitation at 495 nm and detection at 510–550 nm. For observation of vascular bundles, infected leaf segments collected at 6 dpi were cut into small pieces and fixed with 4% glutaraldehyde. The segments were embedded in epoxy resin and sliced with a microtome (0.4 µm thickness). Cross-sections were stained with 1% (wt/vol) Safranin O (Sigma-Aldrich) for 2–3 min and observed by light microscopy.

## Accession numbers

*U. maydis* genes and encoding protein sequences are available at the MIPS database (http://mips. helmholtz-muenchen.de/genre/proj/ustilago/); *tin2*, um05302; *ppi*, um03726. For *Z. mays* genes, sequence data can be found at MaizeSequence.org (http://www.maizesequence.org) under accession numbers; ZmGAPDH, GRMZM2G046804; ZmTTK1, GRMZM2G448633; ZmR1, GRMZM5G822829; ZmC1, GRMZM2G005066; ZmPAL, GRMZM2G074604; ZmCHS, GRMZM2G422750; ZmCHI, GRMZM2G155329; ZmF3H, GRMZM2G062396; ZmDFR, GRMZM2G026930; ZmANS, GRMZM2G345717; ZmBx1, GRMZM2G085381; ZmBx2, GRMZM2G085661; ZmBx5, GRMZM2G063756; ZmBx8, GRMZM2G085054; Zm4CL, GRMZM2G075333, GRMZM2G054013; ZmCCR, GRMZM2G131205, GRMZM2G131836; ZmCAD, GRMZM2G046070, GRMZM5G844562, GRMZM2G167613, AC234163.1_ FGP002, GRMZM2G700188, GRMZM2G118610.

## Acknowledgements

We are very grateful to V Walbot, E Grotewold, E Coe, HK Dooner, MD McMullen, A Polle, and A Gierl for helpful suggestions, and K Hahlbrock for comments on the manuscript. We thank the Maize Genetics Stock Center for providing maize mutant seeds, U Lahrmann for support with microarray data analysis, K Bolte for preparing leaf cross-sections, P Meyer for help with lignin and anthocyanin analysis and V Vincon for assistance with virulence assays. This work was supported by the Max Planck Society, by the DFG Collaborative Research Center 593 and the LOEWE program of the state of Hesse. ST was supported by a postdoctoral research fellowship from the Alexander von Humboldt foundation. IF was supported by the DFG excellence initiative FL3 INST 186/822-1. WV acknowledges funding from the Consortium for Plant Biotechnology Research, Inc. (DOE Prime Agreement No. DEFG36-02GO12026) and the USDA Biomass Research & Development Initiative grant no. 2011–10006–30358.

## Additional information

### Funding

| Funder | Grant reference number | Author |
|---|---|---|
| Max Planck Society | | Shigeyuki Tanaka, Thomas Brefort, Nina Neidig, Armin Djamei, Jörg Kahnt, Regine Kahmann |
| DFG Collaborative Research Center 593 | | Thomas Brefort, Armin Djamei, Regine Kahmann |
| LOEWE Program of the State of Hesse | | Regine Kahmann |
| Alexander von Humboldt Foundation | | Shigeyuki Tanaka |
| DFG Excellence Initiative | FL3 INST 186/822-1 | Ivo Feussner |
| Consortium for Plant Biotechnology Research Inc. | DEFG36-02GO12026 | Wilfred Vermerris |
| USDA Biomass Research & Development Initiative | 2011-10006-30358 | Wilfred Vermerris |

The funders had no role in study design, data collection and interpretation, or the decision to submit the work for publication.

### Author contributions

ST, Designed and performed experiments, Performed the microarray analysis, wrote the manuscript with input from all co-authors; TB, Performed the initial yeast two hybrid screening that identified ZmTTK1; NN, Performed the microarray analysis; AD, Contributed to the transient expression assays and sampling for anthocyanin and lignin analysis; JK, Analyzed the anthocyanins; WV, Generated and provided the maize *bm* mutants; SK, Performed the lignin analysis of infected leaves; KF, Analyzed

the anthocyanins, Analyzed the flavonoids and DIBOA-glucoside of infected leaves; IF, Analyzed the flavonoids and DIBOA-glucoside of infected leaves, Performed the lignin analysis of infected leaves; RK, Directed the project, Wrote the manuscript with input from all co-authors

## Additional files

### Supplementary files

• Supplementary file 1. Plasmids used in this study.

• Supplementary file 2. Primers used in this study.

• Supplementary file 3. *Ustilago maydis* strains used in this study.

• Supplementary file 4. *Saccharomyces cerevisiae* strains used in this study.

### Major dataset

The following dataset was generated:

| Author(s) | Year | Dataset title | Dataset ID and/or URL | Database, license, and accessibility information |
|---|---|---|---|---|
| Kahmann R | 2013 | Maize gene expression after infection of Ustilago maydis SG200 and SG200Dtin2 | http://www.ncbi.nlm.nih.gov/geo/query/acc.cgi?acc=GSE48536 | In the public domain at GEO: http://www.ncbi.nlm.nih.gov/geo/. |

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
