## [Decision Letter]

Thank you for sending your work entitled “Fungal effector exploits anthocyanin induction to prevent lignification” for consideration at *eLife*. Your article has been evaluated by a Senior Editor and 2 reviewers, one of whom is a member of our Board of Reviewing Editors.

The Reviewing editor and the other reviewer discussed their comments before we reached this decision, and the Reviewing editor has assembled the following comments. While the manuscript in its current form is not acceptable for *eLife*, we believe that the work has potential. The reviewers had several serious concerns, and would consider a substantially revised manuscript.

The work utilizes a fungal effector to identify a new interaction with a novel regulator of the phenylpropanoid network that appears to be limited to the monocots. The most serious concern was that, while there is evidence for an anthocyanin role in defense, there was not very good support for a flux trade off argument with respect to the role of lignin, especially since anthocyanins can act directly as antimicrobial compounds of the phytoalexin class in other species, without invoking a lignin link. The flux trade off is highly speculative. One way to support the model would be to determine whether high anthocyanin lines have lowered lignin content. Alternatively, you should merely bring up the lignin link as a speculative hypothesis in the Discussion.

Additional concerns:

1) There were key experimental details mentioned in each specific review (by Reviewer 1 and Reviewer 2, below) that should be included or clarified in the text. These are essential such as with regards to information about the genetic background and phylogeny to provide the necessary assurance for the interpretations being put forward.

2) As the data were presented as regulating pathways and there is extant transcriptome information, it would make sense to provide pathway level views of gene expression at all phenolic genes both phenylpropanoid and flavonoid pathways. It was felt that this pathway level view would help the reader to better place the observations in their broader network context.

*Reviewer #1 comments*:

The authors focus on an effector from *Ustilago maydis* that interacts with a maize protein kinase that is responsible for controlling phenolic metabolism. In this work the authors make the identification of a putative monocot specific phenolic regulator that interacts with the Tin2 effector. This then links to transcriptional control of the phenolic metabolic pathways and is proposed to control a flux based defense system. The molecular analysis of the effector is very strong and detailed. However, I would argue that the novelty is not in the identification that an effector does something but instead in the analysis of phenolic metabolism. It is here that there needs to be work to bring the metabolic interpretation to the same strength as the molecular analysis of the effector.

I think the authors have really over-interpreted the flux redirection concept. This begins with the title that makes an explicit assumption of flux redirection and continues throughout. They have some evidence on the anthocyanins via the CHS mutant but don't have any evidence on the lignification theory. This is exemplified by their final sentence “We suggest that examples where biotic stress leads to anthocyanin induction should be revisited as the deployment of specific translocated effectors (like Tin2) to achieve metabolic rewiring may be a common strategy adopted by plant pathogenic microbes.” Their data definitely shows that the effector rewires but it does not show that this rewiring is key. Given the data, it is completely possible that the anthocyanins are playing a direct role in the infection, which is in agreement with the other citations. To make the rewiring/flux arguments would require mutants in CAD or other lignin steps to show that these have the expected consequences.

To experimentally make the flux theory, it would require absolute quantification of the anthocyanins to show that this is a comparable amount of carbon going into anthocyanins that is lost from the phenolics. The anthocyanins are currently not quantified.

Figure 6, how did the authors define orthologs for TTK1? A key point of the manuscript is that this is a unique phenolic regulator for these monocots but there is no description of how the authors ruled out dicot orthologues? Is this part of a larger gene family? This material must be significantly expanded.

Figure 11, the authors should provide the specific gene identifier from the maize genome for each gene as there is more than one copy of a number of these genes. The NP nomenclature isn't specifically linked to the maize genome sequence.

In Figure 12, the three different SG200dtin2 strains should be analyzed collectively using a nested anova where the independent isolates are nested within the broader genotype to ensure that there is really a difference. Currently only one of the three is different at a standard level of significance (<0.05). The other two are by definition (>0.05) not significant. As the increase in phenolics is a key assumption of the paper this data needs to be strongly supported and currently it is not as solid as it should be.

*Reviewer #2 comments*:

This manuscript describes a possible mechanism by which the maize fungus *Ustilago maydis* induces accumulation of anthocyanins as a mechanism to prevent lignin formation. Anthocyanin induction would be accomplished by induction of the Tin2 protein, which stabilizes a kinase (ZmTTK1), which in turn affects somehow anthocyanin formation. While the model presented is novel and of significant potential interest, the manuscript lacks some important controls that make it difficult to evaluate the validity of the conclusions.

1) Figure 1 needs a much better description in the Results section. The spectra in Figure 1 are very difficult to see and it is very difficult to compare the replicates (which as far as I can tell, look very different in terms of the absorption and MS spectra – why?)

2) The authors refer to Figure 3 for anthocyanin gene induction, but this is just a hierarchical cluster analysis with no gene information. It would be more useful to show the expression level of the anthocyanins genes in the various treatments and replicates.

3) ‘tin2 mutant phenotype’ paragraph of the Results: I can't follow the rationale of what the chelating activity of flavonoids has to do with the observations.

4) More worrisome is that in order to form anthocyanins, the entire pathway should be induced, not just 2 genes.

5) There are no error bars in Figure 4. 

6) It is unclear if a library was used for the yeast two-hybrid screens - I could find no details in the M&M.

7) Figure 8 is lacking a negative control.

8) The immunoprecipitate in Figure 9 needs to be separated in a lower percentage PAGE gel to visualize the change in molecular weight and the formation of a single or multiple ubiquitinated forms.

9) How do the authors explain the cytoplasmic localization of mCherry-ZmR1, if the Wessler/Raikhel labs showed that it has multiple NLSs (Shieh et al., 1993)? In their experiments, Tin2 is unlikely to have been present.

10) Where the c2 mutants in the same genetic background as the controls? There is a huge genetic diversity between maize lines and all the differences could be attributed to phenomena unlinked to anthocyanins. According to the M&M, most of the infection experiments were done in hybrid maize, while the mutants used are in various (and often mixed) genetic backgrounds.

11) Is the amount of anthocyanin formed sufficient to explain a competition with the lignin pathway?

12) Discussion: It is incorrect that there is no anthocyanins in maize seedlings – anthocyanins accumulate at significant levels in most inbred lines in the coleoptile, leaf margins and often stem (not to mention later adventitious roots).

---

## [Author Response]

*The work utilizes a fungal effector to identify a new interaction with a novel regulator of the phenylpropanoid network that appears to be limited to the monocots. The most serious concern was that, while there is evidence for an anthocyanin role in defense, there was not very good support for a flux trade off argument with respect to the role of lignin, especially since anthocyanins can act directly as antimicrobial compounds of the phytoalexin class in other species, without invoking a lignin link. The flux trade off is highly speculative. One way to support the model would be to determine whether high anthocyanin lines have lowered lignin content. Alternatively, you should merely bring up the lignin link as a speculative hypothesis in the Discussion*.

We agree that the link to lignin is speculative and we have made sure throughout the manuscript that this is a hypothesis that remains to be proven by additional experiments.

However, we now include data where we have tested the available maize brown-midrib mutants that are affected in lignin biosynthesis. This data has revealed that with the exception of the bm2 mutant all other mutants display significantly enhanced disease symptoms of *U. maydis*. In addition tumor formation, which is largely locally confined after infection of wild type maize, is spreading to much larger leaf areas in the bm mutants. This illustrates independently that lignification does limit *U. maydis* symptoms.

*Additional concerns*:

*1) There were key experimental details mentioned in each specific review that should be included or clarified in the text. These are essential such as with regards to information about the genetic background and phylogeny to provide the necessary assurance for the interpretations being put forward*.

We have carefully revised the manuscript and have addressed all of these points.

*2) As the data were presented as regulating pathways and there is extant transcriptome information, it would make sense to provide pathway level views of gene expression at all phenolic genes both phenylpropanoid and flavonoid pathways. It was felt that this pathway level view would help the reader to better place the observations in their broader network context*.

We have now included pathway level transcriptomic information (qPCR data) for the entire anthocyanin pathway, the lignin pathway and the DIMBOA biosynthesis pathway (as we picked this up as an additional pathway possibly affected by Tin2).

Reviewer #1 comments:

*The authors focus on an effector from* Ustilago maydis *that interacts with a maize protein kinase that is responsible for controlling phenolic metabolism. In this work the authors make the identification of a putative monocot specific phenolic regulator that interacts with the Tin2 effector. This then links to transcriptional control of the phenolic metabolic pathways and is proposed to control a flux based defense system. The molecular analysis of the effector is very strong and detailed. However, I would argue that the novelty is not in the identification that an effector does something but instead in the analysis of phenolic metabolism. It is here that there needs to be work to bring the metabolic interpretation to the same strength as the molecular analysis of the effector*.

We are very much in disagreement with the assessment that “ the novelty is not in the identification that an effector does something but instead in the analysis of phenolic metabolism”. This reviewer is obviously not fully aware of the fact that there are hardly any reports on the function of novel fungal or oomycete effectors that are not Avr proteins. Therefore, we feel somewhat disappointed that the part on elucidation how Tin2 functions has not been met with more appreciation.

In the meantime, however, we have done whatever possible to strengthen the link with lignin (see below for more details) with the most important result being that with one exception all maize mutants affected in lignin biosynthesis display significantly enhanced virulence of *U. maydis* and in addition tumor formation which is locally confined in wild type maize is spreading to much larger leaf areas. This illustrates via a completely independent approach that lignification does limit *U. maydis* symptoms.

*I think the authors have really over-interpreted the flux redirection concept. This begins with the title that makes an explicit assumption of flux redirection and continues throughout. They have some evidence on the anthocyanins via the CHS mutant but don't have any evidence on the lignification theory. This is exemplified by their final sentence “We suggest that examples where biotic stress leads to anthocyanin induction should be revisited as the deployment of specific translocated effectors (like Tin2) to achieve metabolic rewiring may be a common strategy adopted by plant pathogenic microbes.” Their data definitely shows that the effector rewires but it does not show that this rewiring is key. Given the data, it is completely possible that the anthocyanins are playing a direct role in the infection, which is in agreement with the other citations. To make the rewiring/flux arguments would require mutants in CAD or other lignin steps to show that these have the expected consequences*.

We have now rephrased all sentences where we proposed a flux redirection by Tin2 (including the title) and have been very careful in pointing out what we have actually shown and what is our model. Unfortunately, we could not yet test whether the purple maize mutants that show anthocyanin formation constitutively have the expected lower lignin content. The reason for this is that the plants display the purple phenotype only when almost fully grown under natural light conditions. The mutant plants we are growing in our greenhouse since late October, unfortunately, do not display the purple leaf phenotype yet. Therefore, we could not test the effects of anthocyanin production on lignin content in leaves. However, based on our newly included data that show enhanced virulence of *U. maydis* in several lignin mutants of maize, which was the expected consequence, we feel more convinced than before that our ideas were correct.

*To experimentally make the flux theory, it would require absolute quantification of the anthocyanins to show that this is a comparable amount of carbon going into anthocyanins that is lost from the phenolics. The anthocyanins are currently not quantified*.

We have now provided the quantification of anthocyanins but were unable to do this for the lignins for technical reasons, i.e., insufficient amounts of material and lack of internal standards for the extraction efficiency. In addition, we have noted the upregulation of several compounds diverting from the anthocyanin pathway as well as of DIMBOA, which makes a flux assessment more difficult. We have discussed all this data in the context of our hypothetical model.

Figure 6*, how did the authors define orthologs for TTK1? A key point of the manuscript is that this is a unique phenolic regulator for these monocots but there is no description of how the authors ruled out dicot orthologues? Is this part of a larger gene family? This material must be significantly expanded*.

We are very thankful for this comment. By lowering the stringency we detected two putative ZmTTK1 paralogs in maize and one putative ortholog in *A. thaliana*. This is now included in the results and in part in Figure 7.

Figure 11*, the authors should provide the specific gene identifier from the maize genome for each gene as there is more than one copy of a number of these genes. The NP nomenclature isn't specifically linked to the maize genome sequence*.

This has now been included in Material and methods and in the respective figures depicting qPCR results.

*In*
Figure 12*, the three different SG200dtin2 strains should be analyzed collectively using a nested anova where the independent isolates are nested within the broader genotype to ensure that there is really a difference. Currently only one of the three is different at a standard level of significance (<0.05). The other two are by definition (>0.05) not significant. As the increase in phenolics is a key assumption of the paper this data needs to be strongly supported and currently it is not as solid as it should be*.

We have repeated the experiment and have observed a similar tendency of the results, i.e., a higher lignin content in maize tissue infected with the tin2 mutant compared to wild type. However, in the statistical analysis (t-test) a significant difference is only detected for one of the three tested tin2 mutants. The nested annova gave the same result. Since we have used the t-test in all other analyses, we decided to show the t-test results also here. In addition, we have now more extensively discussed why we may not have detected a stronger difference and explain this with the enhanced lignification in the vascular tissue only

Reviewer #2 comments:

*This manuscript describes a possible mechanism by which the maize fungus* Ustilago maydis *induces accumulation of anthocyanins as a mechanism to prevent lignin formation. Anthocyanin induction would be accomplished by induction of the Tin2 protein, which stabilizes a kinase (ZmTTK1), which in turn affects somehow anthocyanin formation. While the model presented is novel and of significant potential interest, the manuscript lacks some important controls that make it difficult to evaluate the validity of the conclusions*.

*1)*
Figure 1
*needs a much better description in the Results section. The spectra in*
Figure 1
*are very difficult to see and it is very difficult to compare the replicates (which as far as I can tell, look very different in terms of the absorption and MS spectra – why?*)

We have redone the analysis and have now quantified the different anthocyanins. The Figure 4 was completely redone and now represents absolute amounts in a bar-chart representation.

*2) The authors refer to*
Figure 3
*for anthocyanin gene induction, but this is just a hierarchical cluster analysis with no gene information. It would be more useful to show the expression level of the anthocyanins genes in the various treatments and replicates*.

We now provide the expression analysis of all anthocyanin biosynthesis genes by qPCR in Figure 3.

*3) ‘tin2 mutant phenotype’ paragraph of the Results: I can't follow the rationale of what the chelating activity of flavonoids has to do with the observations*.

We have deleted this.

*4) More worrisome is that in order to form anthocyanins, the entire pathway should be induced, not just 2 genes*.

We have now analyzed the expression of all genes and show that without exception they are all induced.

*5) There are no error bars in*
Figure 4.

Error bars are now included.

*6) It is unclear if a library was used for the yeast two-hybrid screens - I could find no details in the M&M*.

We apologize for omitting this and have included a detailed description now in the Materials and methods section.

*7)*
Figure 8
*is lacking a negative control*.

We had included Tin2_26-202_, a mutant protein lacking biological activity, and this protein shows less interaction with ZmTTK1. This, in our view, constitutes the best control possible.

*8) The immunoprecipitate in*
Figure 9
*needs to be separated in a lower percentage PAGE gel to visualize the change in molecular weight and the formation of a single or multiple ubiquitinated forms*.

We are not interested in figuring out whether the ubiquitinated species seen contains a single or multiple ubiquitin chains. For us, it was important to show that there is ubiquitination and this is absent when the two serine residues in the phosphodegron motif are substituted with alanine. Therefore, we have not redone this experiment.

*9) How do the authors explain the cytoplasmic localization of mCherry-ZmR1, if the Wessler/Raikhel labs showed that it has multiple NLSs (Shieh et al., 1993)? In their experiments, Tin2 is unlikely to have been present*.

In Shieh et al., (1993), the authors use onion cells for localization studies. We consider it conceivable, that NLS motifs are masked in the homologous maize system that we have used. Alternatively, ZmR1 may be generally unstable when localized in the cytosol of maize without Tin2. The fact that we detect ZmR1 in the cytosol without Tin2 in a few cells, may depend on higher expression levels in those cells. The number of molecules introduced cannot be controlled precisely when using biolistic bombardment for transient expression.

*10) Where the c2 mutants in the same genetic background as the controls? There is a huge genetic diversity between maize lines and all the differences could be attributed to phenomena unlinked to anthocyanins. According to the M&M, most of the infection experiments were done in hybrid maize, while the mutants used are in various (and often mixed) genetic backgrounds*.

C2 (used as control) and c2 (lacking chalcone synthase) are near isogenic lines which we now mention in Materials and Methods.

*11) Is the amount of anthocyanin formed sufficient to explain a competition with the lignin pathway*?

Unfortunately, we cannot provide an answer to this question. We have now quantified anthocyanins but were unable to do this for the lignins for technical reasons, i.e., insufficient amounts of material and lack of internal standards for the extraction efficiency. In addition, we have noted the upregulation of several compounds diverting from the anthocyanin pathway as well as of DIMBOA, which makes a flux assessment more difficult. We have eliminated all sentences implying such a competition in the Results and Discussion and refer to this possibility only in our model where it is clearly stated that this is a hypothetical model.

*12) Discussion: It is incorrect that there is no anthocyanins in maize seedlings – anthocyanins accumulate at significant levels in most inbred lines in the coleoptile, leaf margins and often stem (not to mention later adventitious roots)*.

We have now corrected this and we have included additional references.